# MoQE: Improve Quantization Model performance via Mixture of Quantization Experts.

## Abstract

Quantization method plays a crucial role in improving model efficiency and reducing deployment costs, enabling the widespread application of deep learning models on resource-constrained devices. However, the quantization process inevitably introduces accuracy degradation. In this paper, we propose Mixture of Quantization Experts( abbr. MoQE), a quantization inference framework based on the Mixture-of-Experts (MoE) architecture, aiming to jointly improve the performance of quantization models. MoQE combines multiple quantization variants of one full-precision model as specialized "quantization experts" and dynamically routes input data to the most suitable expert based on its characteristics. MoQE alleviates the performance degradation commonly seen in single quantization models through specialization quantization expert models. We design lightweight, structure-aware router models tailored for both CV and NLP tasks. Experimental evaluations on ResNet, LLaMA, and Qwen model families across benchmark datasets including ImageNet, WikiText, C4, and OpenWebText demonstrate that MoQE achieves performance comparable to SOTA quantization model, without incurring significant increases in inference latency.

## 1 Introduction

Quantization method plays a pivotal role in the field of machine learning, particularly in enhancing model efficiency and reducing resource consumption. As deep learning models grow increasingly complex, their demand for computational resources escalates, constraining deployment on resource-limited devices and increasing operational costs. Quantization mitigates storage requirements and computational complexity by reducing the precision of model weights and activations. Furthermore, quantization method streamlines the model optimization pipeline, enabling developers to achieve efficient deployment within shorter timeframes and accelerating time-to-market for AI-driven products. Consequently, quantization method serves not only as a critical enabler for improving the accessibility and practicality of machine learning models but also as a key facilitator in the broader dissemination of artificial intelligence technologies.

Quantization techniques exhibit notable advantages in compressing deep neural network models and minimizing computational overhead. However, their practical deployment faces several critical challenges. Quantization inherently involves mapping high-precision floating-point parameters to low-bit representations, a process that inevitably introduces information loss, thereby degrading model generalization and accuracy. This degradation is particularly pronounced under ultralow bit-width settings (e.g., below 8 bits), where quantization noise can significantly impair representational capacity, leading to performance deterioration, especially in complex tasks or applications with stringent accuracy requirements. Thus, achieving high efficiency without compromising model performance remains a central research challenge.

The MoE (Mixture of Experts) system is an artificial intelligence architecture based on a combination of expert models, designed to solve complex problems by integrating the capabilities of multiple specialized models (the "experts"). Each expert handles a specific aspect of the input data, while a component called the "gating network" determines how to allocate tasks to different experts based on the input's characteristics. In this paper, we named the gating network as a router model. This approach allows the model to scale to very large sizes while maintaining computational efficiency, as not all experts need to compute every input. MoE systems are particularly suitable for applications

requiring high flexibility and strong expressive power, such as NLP and image recognition, delivering more accurate and personalized services and predictions. As technology advances, MoE has become a key technique for building efficient, large-scale machine learning models. However, there are few works which focus on how to improve quantization model performance via MoE system.

To address the common industrial constraint that only a single pretrained model is available for quantization, we propose the Mixture of Quantization Experts (MoQE), an MoE-style inference system that improves accuracy while keeping inference latency comparable to a single quantization model. Prior studies Liu et al. (2025) indicate that different quantization variants of the same full-precision model exhibit heterogeneous performance degradation across distinct sub-datasets. Motivated by this observation, MoQE assigns different sub-datasets to specialized quantization models—termed "quantization experts"—thereby improving the overall system performance. Notably, as the number of quantization experts increases, the coverage and representational diversity of the system expand, leading to progressive performance gains.

To facilitate dynamic routing of input data to appropriate quantization experts, we design two router models tailored respectively for CV and NLP models. These routers leverage key structural components from their respective base models, ensuring compatibility and semantic relevance, while maintaining a significantly smaller scale to ensure rapid execution with minimal computational overhead.

We evaluate MoQE using ResNet, LLaMA, and Qwen series models on benchmark datasets including ImageNet, WikiText, C4, and OpenWebText. Experimental results demonstrate that MoQE consistently achieves performance comparable to that of the best individual quantization model, without incurring significant increases in inference latency.

Our contributions are listed as the following:

1.We design a Mixture of Quantization Experts system, abbr. MoQE. By integrating quantization models into an expert system, MoQE routes different data subsets to the quantization models where they perform best, resulting in overall system performance superior to that of any single expert.

2.We design data router models for both CV and NLP tasks. The router model is a lightweight classification model with a short inference time, achieving inference speed of MoQE system comparable to that of a single quantization model.

3.Through experiments, we have validated the performance of MoQE on both CV and NLP tasks. Experimental results show that the performance of the MoQE system exceeds that of any individual quantization expert model comprising it. Furthermore, the performance of the MoQE system improves as the number of expert models increases. Experiments code address is https://github.com/paper-submission-goii/MoQE.

## 2 RELATED WORK

**Quantization** falls into quantization-aware training (QAT) and post-training quantization (PTQ), with PTQ favored for deployment due to low cost. HAWQ-v2 Dong et al. (2019) uses Hessian trace to measure layer sensitivity, while Zhe et al. (2019) formulates mixed-precision allocation as Lagrangian optimization. AdaQuant Hubara et al. (2020) minimizes per-layer quantization error on calibration data. Recent works Liu et al. (2023); Shang et al. (2024) improve PTQ via activation distribution adjustment and mutual information, but still struggle at very low bits and require calibration. Our method avoids fine-tuning and calibration, achieving strong performance and efficiency in ultralow-bit regimes. **Mixture of Experts(MoE)** Jacobs et al. (1991); Eigen et al. (2013) employs multiple experts with routing mechanisms. Hard routing assigns experts to predefined modalities Bao et al. (2022); Long et al. (2023), simplifying deployment without learned routers. In contrast, soft routing in NLP Shazeer et al. (2017); Fedus et al. (2022) enables dynamic, probabilistic token-expert assignment for sparsity and scalability. Soft-routed MoEs are also adapted to multimodal models like EVE Chen et al. (2024) and LIMoE Mustafa et al. (2022). Our work focuses on soft routing for flexible, data-driven expert selection.

# 3 METHODOLOGY

## 3.1 MIXTURE OF QUANTIZATION EXPERTS

To mitigate the degradation of model accuracy caused by quantization, this paper proposes a novel framework termed Mixture of Quantization Experts (MoQE), the working pattern of MoQE is shown in Figure 1. In the MoQE architecture, multiple quantization variants of the same full-precision model are integrated through an expert system paradigm. These quantization models are typically derived via different quantization methods on the same full-precision model. The selection of which specific quantization model to activate is governed by a dedicated router model. The design and formulation of the router model will be detailed in the following section. The activated quantization expert model performs the inference on this sample. The router is trained on a training dataset.

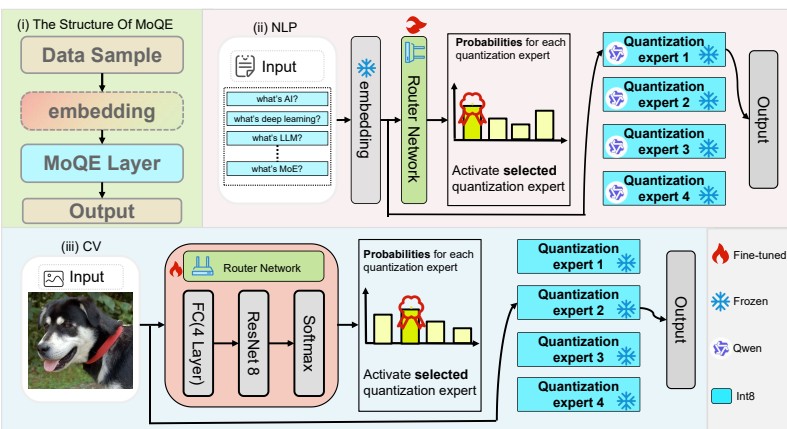

Figure 1: The working pattern of MoQE. The data sample first passes through a router model which may involve an embedding step, to determine which quantization expert model to activate, and then is fed into the corresponding quantization expert model for inference.

The theoretical foundation of this approach stems from the observation that different quantization models exhibit distinct biases (unfairness) across various regions of the input data space Liu et al. (2025). Specifically, for certain subsets of data, one quantization model may consistently yield lower prediction errors (i.e., lower loss values) than others. For example, as shown in the Figure 2(a), we evaluate the perplexity of five mainstream quantization methods—GPTQ, SmoothQuant, K-Quants, imatrix, and AWQ on seven different sub-datasets (Subdata1–7) of the Qwen 1.7B model. The results show that there are significant differences in performance among these subsets. For instance, SmoothQuant has the lowest perplexity on Subdata4, while AWQ performs the best on Subdata2.

By leveraging the router model to dynamically assign samples to the most suitable expert, MoQE effectively selects, at each inference step, the model with the lowest loss function value on this subset of data. As illustrated in the Figure 2(b), this strategy enables the system to synthesize a composite decision boundary that outperforms any individual quantization model. Consequently, the router model can be interpreted as a multi-class classifier, and its effective design is critical to ensuring that the overall system performance exceeds that of any constituent model.

It is important to note that the router model operates on vectorized representations of input data, implying that raw data formats may first be transformed into appropriate embedding. This process is optional depending on the task. For example, in CV tasks, images are naturally represented as tensors and can be directly processed by the router model. Thus, CV tasks do not require embedding process. In contrast, for NLP tasks, textual inputs must be converted into dense vector representations—typically through embedding techniques—before they can be handled. The quality of this embedding process directly influences the router model's classification accuracy.

In the context of NLP, MoQE uses the pre-existing embedding layer of the original full-precision model as its embedding layer. This choice serves two purposes: (1) it avoids the need to train a separate embedding layer for routing, which would be computationally expensive and potentially

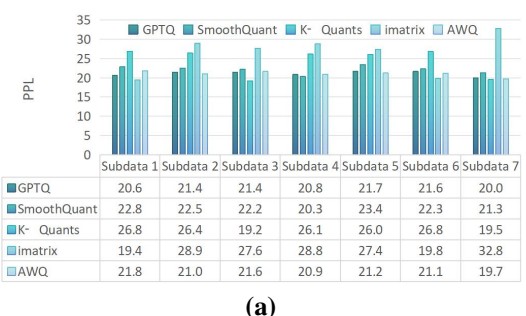 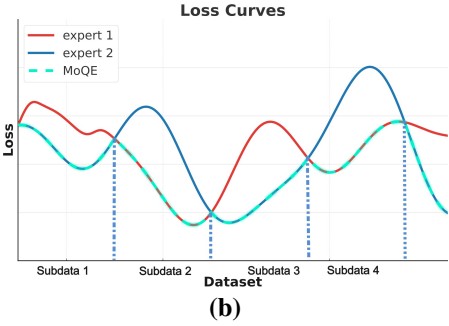

|  | Subdata 1 | Subdata 2 | Subdata 3 | Subdata 4 | Subdata 5 | Subdata 6 | Subdata 7 |
|---|---|---|---|---|---|---|---|
| GPTQ | 20.6 | 21.4 | 21.4 | 20.8 | 21.7 | 21.6 | 20.0 |
| SmoothQuant | 22.8 | 22.5 | 22.2 | 20.3 | 23.4 | 22.3 | 21.3 |
| K- Quants | 26.8 | 26.4 | 19.2 | 26.1 | 26.0 | 26.8 | 19.5 |
| imatrix | 19.4 | 28.9 | 27.6 | 28.8 | 27.4 | 19.8 | 32.8 |
| AWQ | 21.8 | 21.0 | 21.6 | 20.9 | 21.2 | 21.1 | 19.7 |

(a)                (b)

Figure 2: (a) The specific performance of multiple quantization methods for Qwen 1.7B on different sub-datasets (Subdata1 - 7) (b) The explanations of MoQE system which consists of two quantization experts. MoQE always selects the model with the lower loss on this subset of data.

degrade performance due to misalignment; and (2) it reduces inference cost because using the same embedding layer avoids an extra embedding pass. Therefore, the quantization experts' embedding layer is left in full precision.

In our experimental design, we primarily focus on quantization variants of a single full-precision model, reflecting typical industrial deployment scenarios where only one full-precision model is available for quantization. Moreover, in NLP tasks, the shared embedding layer constraint limits the framework to experts originating from the same base model, which implies that all quantization experts come from the same full-precision base model. However, the MoQE framework is, in principle, extensible to quantization models derived from multiple distinct full-precision models. Prior work Liu et al. (2025) suggests that for well-trained models, performance variations across quantization versions in different data regions are largely determined by local gradient and Hessian properties.

### 3.2 ROUTER MODEL

The design of router models must be tailored to the intrinsic characteristics of different task domains. For instance, NLP tasks necessitate a strong emphasis on contextual dependencies, while CV tasks primarily rely on spatial and hierarchical visual features. By carefully calibrating the architectural and parametric configurations of the router model, classification accuracy can be significantly improved, ensuring that each quantization expert model operates at peak efficacy within its targeted data distribution. This, in turn, contributes to enhanced overall inference accuracy. To this end, and to address the prevalent use cases in modern deep learning, we have designed two specialized router models tailored specifically for CV and NLP tasks, respectively—each optimized to exploit the structural priors inherent in its respective domain. This section details the router model's design and implementation. Convergence and error analyses are provided in appendix A.7.

**CV Router Model**

We propose a lightweight, multi-head SEResNet-8 router architecture. The processing pipeline commences with a three-layer MLP tasked with the non-linear transformation of the input global features. Each linear transformation is followed by Batch Normalization (BN) and a ReLU activation. Subsequently, a three-stage SEResNet-8 module, configured with channel dimensions of 16, 32, and 64 and a stride pattern of 1→2→2, is employed to extract hierarchical local visual representations. Finally, to generate the routing logits, the resulting feature map is flattened into a sequence and fed into an 8-head self-attention module. This design is intended to aggregate global information, thereby facilitating an expert assignment strategy that is both globally-aware and load-balanced, all while ensuring minimal computational overhead.

**NLP Router Model** The core design principle of our NLP router is to endow router model decisions with the same level of semantic richness and contextual sensitivity as the expert computations.The structure of router model is shown in Figure 3. The foundation of our architecture is a Transformer encoder. Its primary task is to process the entire input sequence in parallel, transforming each static token embedding into a context-aware representation. This initial stage captures the global linguis-

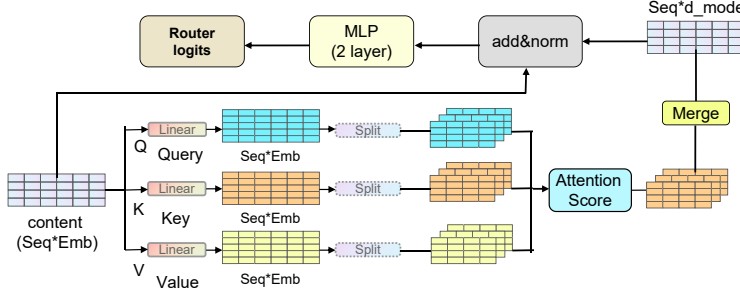

Figure 3: Structure of the router model

tic environment surrounding each token. Building upon these general-purpose contextual vectors, a dedicated self-attention module subsequently performs a function of focused refinement. It distills and re-weights the existing representations to identify the most critical contextual cues, thereby optimizing the routing decision. This role forms a crucial complement to the general-purpose semantic understanding provided by the initial encoder. Finally, MLP maps the refined, context-rich feature vectors to the final routing logits. It employs a non-linear projection to translate these complex representations into definitive scores for each expert.

### 3.3 ROUTER FINE-TUNING

In the fine-tuning stage of the MoQE system, the core objective is to train an efficient and stable router model that can dynamically assign input data to the most suitable quantization expert based on its features. To achieve this, a key principle is that the weights of all quantization experts remain completely frozen throughout the training process. The router model itself is maintained at full-precision during training to ensure it possesses maximum representational capacity for making accurate routing decisions.

Each sample is fed in parallel to all N frozen quantization experts, and the prediction loss is calculated for each expert. The expert that yields the lowest loss is designated as the "optimal expert" for that sample, serving as the target label for the router model to learn. We designed a composite loss function to ensure that the router model not only learns to assign samples accurately but also avoids collapsing into a state where it relies on only a few experts. The total objective function consists of two components:

$$\mathcal{L} = \mathcal{L}_{\text{CE}} + \alpha_{\text{dyn}}\mathcal{L}_{\text{bal}}, \mathcal{L}_{\text{bal}} = N \cdot \sum_{i=1}^{N} P_i \cdot F_i \tag{1}$$

$$P_i \triangleq \mathbb{E}_{x \sim \mathcal{B}}[p_i(x)] \approx \frac{1}{B} \sum_{x \in \mathcal{B}} p_i(x), F_i \triangleq \frac{n_i}{B} \tag{2}$$

Here, $P_i$ is the mean routing probability for expert $i$ and $F_i$: the fraction of samples actually dispatched to expert $i$. $N$ denotes the number of experts, $p_i(x)$ is the softmax probability of routing sample $x$ to expert $i$, $mathcalB$ is the current mini-batch, $n_i$ is the number of samples in the batch actually dispatched to expert $i$. The influence of this balancing regularizer is controlled by a dynamic weight, $\alpha_{\text{dyn}} = \alpha(1 + \sigma_t)$, $t$ is the epoch number. Here, $\alpha$ is a hyper-parameter with an initial value of 0.02 , while $\sigma_t$ is the relative standard deviation of expert usage frequencies. We linearly decay $\alpha$ to 0 in the final stage of training, thereby shifting the optimization focus toward minimizing $\mathcal{L}_{\text{CE}}$ and stabilizing the routing policy while preserving system diversity. Detailed training configurations are described in Appendix A.1.

### 3.4 THEORETICAL ANALYSIS

In this section, we analyze the theoretical bound of the proposed MoQE framework.

**Theorem 1** *Let $\mathcal{A}$ denote the set of data subsets. For any subset $a \in \mathcal{A}$, let $L_k(a)$ be the loss of the k-th quantization expert. Let $\mathbb{L}_{Single}$ represent the total loss of any single quantization expert , and*

$\mathbb{L}_{MoQE}$ *represent the total loss of the MoQE system utilizing an optimal routing strategy. Then, the following inequality strictly holds:*

$$\mathbb{L}_{MoQE} \leq \mathbb{L}_{Single} \tag{3}$$

for every subset $a \in \mathcal{A}$:

$$\arg \min_{k \in \{1,...,K\}} L_k(a) = fixed \tag{4}$$

The performance of MoQE is equivalent to that of that expert. We provide the detailed formal proof and error bound derivation in Appendix A.3.

## 4 EXPERIMENTS

The experiments are divided into three parts: NLP, CV, and Configurations experiment(Ablation experiments). In NLP tasks, our main focus is on current mainstream large language models, namely the Qwen and Llama series. In CV tasks, the primary models are those based on residual architectures, specifically the ResNet and MobileNet series. Finally, we will analyze how different system configurations affect performance, which we use as an ablation experiment.

### 4.1 NLP EXPERIMENTS

To ensure a fair comparison, we maintained the embedding layers in full precision (FP16) across all baseline methods (e.g., GPTQ, SmoothQuant), consistent with the MoQE configuration. This aligns with standard practices in existing LLM Post-Training Quantization (PTQ) literature, motivated by three key factors: (1) the inherent sparsity of embeddings limits potential storage compression gains; (2) embedding lookups are memory-bound operations rather than compute-bound, meaning quantization offers negligible arithmetic acceleration; and (3) embeddings require high precision to preserve semantic integrity, making them highly sensitive to quantization errors.

#### 4.1.1 EXPERIMENT CONFIGURATIONS

The experiment configurations are listed as follows:

**Model** We use standard Qwen-0.6B, Qwen-1.7B, Qwen-4B, and Llama-3B as our experimental models. Int8 quantization is applied in this section.

**Router Training Setting** During training of the router model, the Adam optimizer hyperparameter settings are kept consistent across different model scales: the learning rate is set to $5.00 \times 10^{-5}$ for all models, the cosine learning rate scheduler is employed, the batch size is 8, gradient accumulation is applied with 6 steps, and the weight decay coefficient is set to 0.01. Additional training details are provided in the Appendix.

**Benchmarks and MoQE Setting** In our experiments, we use SmoothQuant Xiao et al. (2023), GPTQ Frantar et al. (2022), Imatrix, and K-quantization. To evaluate the impact of the number of experts, we additionally include AWQ Lin et al. (2024) as a benchmark in the configuration (Ablation) experiments. The quantization expert models in MoQE consist of the above-mentioned benchmark models.

We present the performance of MoQE system when the quantization expert models are derived from the same full-precision model.

**Dataset** We use the C4, WikiText-2, and OpenWebText datasets to evaluate our algorithm. C4, a cleaned version of Common Crawl filtered for quality, assesses model robustness on noisy, real-world web text. WikiText-2, composed of structured Wikipedia articles, provides a standard benchmark for evaluating perplexity and modeling long-range dependencies. OpenWebText, a corpus of user-endorsed online content, tests the model's ability to generate fluent, human-preferred text. These datasets are widely used in language modeling benchmark. In the following, we abbreviate WikiText-2 as Wiki and OpenWebText as WebText.

### 4.1.2 EXPERIMENTAL RESULTS ANALYSIS

| PPL | Qwen0.6 | | | Qwen1.7B | | | Llama3B | | | Qwen4B | | |
|---|---|---|---|---|---|---|---|---|---|---|---|---|
| | C4 | WiKi | webtext | C4 | WiKi | webtext | C4 | WiKi | webtext | C4 | WiKi | webtext |
| GPTQ | 25.63 | 21.07 | 19.76 | 19.32 | 16.85 | 14.96 | 16.87 | 14.30 | 13.87 | 19.37 | 14.47 | 13.66 |
| SmoothQuant | 25.94 | 22.11 | 20.34 | 18.96 | 16.73 | 15.08 | 17.69 | 15.17 | 12.13 | 16.71 | 22.45 | 20.97 |
| K-Quants | 32.38 | 24.54 | 21.68 | 21.40 | 17.58 | 16.31 | 17.34 | 15.21 | 15.14 | 21.45 | 17.34 | 15.44 |
| imatrix | 27.84 | 26.38 | 22.45 | 19.24 | 18.54 | 16.38 | 12.18 | 11.82 | 10.93 | 18.32 | 19.89 | 16.38 |
| MoQE(Our) | 24.82 | 20.16 | 18.97 | 18.13 | 15.97 | 14.02 | 11.54 | 11.01 | 10.03 | 15.94 | 13.69 | 12.89 |
| FP16 | 17.18 | 12.76 | 11.32 | 13.48 | 9.53 | 9.02 | 13.56 | 12.46 | 11.57 | 11.84 | 7.95 | 6.35 |

Table 1: Performance Comparison between the MoQE System and Individual Expert Models in NLP Tasks

As shown in Table 1, the experimental results demonstrate the performance of the MoQE system across various large language models (Qwen0.6B, Qwen1.7B, Llama3B, and Qwen4B) and multiple NLP datasets (C4, WikiText-2, and OpenWebText). The evaluation metric is perplexity (lower is better), and MoQE is compared against several state-of-the-art quantization methods, including GPTQ, SmoothQuant, K-Quants, and imatrix, with FP16 serving as the full-precision baseline.

Overall, MoQE consistently achieves the lowest perplexity across nearly all model sizes and datasets, indicating its superior ability to preserve language modeling performance under quantization. Notably, MoQE not only outperforms all other quantization methods but also significantly closes the gap with the full-precision FP16 models. For instance, on the Qwen0.6B model, MoQE achieves perplexities of 24.82 (C4), 20.16 (WikiText-2), and 18.97 (OpenWebText), which are substantially lower than those of the next best method (imatrix) and much closer to the FP16 baseline (17.18, 12.76, 11.32) than any other quantization approach.

This trend persists across larger models. On Qwen1.7B, MoQE obtains 18.13 (C4), 15.97 (WikiText-2), and 14.02 (OpenWebText), again outperforming all competitors and demonstrating strong scalability. In the case of Llama3B and Qwen4B, MoQE achieves the most significant gains—on Llama3B/OpenWebText, MoQE reaches a perplexity of 10.03, surpassing even the FP16 baseline (11.57), which may be attributed to regularization effects induced by the MoE router model. Similarly, on Qwen4B, MoQE achieves 15.94 (C4), 13.69 (WikiText-2), and 12.89 (OpenWebText), significantly outperforming other quantization methods and approaching or exceeding FP16 performance on certain tasks.

These results validate that the MoQE framework effectively combines the strengths of multiple quantization experts through intelligent routing, minimizing quantization-induced degradation. Its consistent superiority across diverse architectures and datasets underscores its robustness and generalizability, making it a highly effective solution for deploying large language models under resource-constrained conditions.

## 4.2 CV EXPERIMENTS

### 4.2.1 EXPERIMENTS CONFIGURATIONS

**Model** We use standard ResNet50, ResNet101, and MobileNetV2 as the experimental models. Int8 quantization is applied in this section.

**Router Training Setting** In training the router model, the same hyperparameter settings for the AdamW optimizer were used across different backbone networks: the learning rate is set to $5.00 \times 10^{-5}$ for all models, the learning rate schedule employs cosine annealing, the batch size was set to 48, gradient accumulation over 2 steps was applied, and weight decay was set to 0. Additional training details are provided in the Appendix.

**Benchmarks and MoQE Setting.** In our experiments, we use BRECQ Li et al. (2021), DSConv Nascimento et al. (2020), N2UQ Liu et al. (2021), and a QAT-trained model. The quantization expert models in MoQE consist of the above-mentioned benchmark models.

**Dataset** We use the ImageNet dataset, which remains a cornerstone in computer vision and is widely recognized as a benchmark for image classification and visual representation learning. It comprises over one million annotated images across 1,000 semantic categories.

### 4.2.2 EXPERIMENTAL RESULTS ANALYSIS

| Acc | ResNet50 | ResNet101 | MobileNetV2 |
|---|---|---|---|
| BRECQ | 76.44% | 77.60% | 69.10% |
| QAT | 77.09% | 77.50% | 71.10% |
| N2UQ | 76.40% | 78.30% | 71.11% |
| DSConv | 76.08% | 77.90% | 70.32% |
| MoQE(Our) | 78.01% | 78.91% | 71.36% |
| FP16 | 80.21% | 83.25% | 71.80% |

Table 2: Performance Comparison between the MoQE System and Individual Expert Models in CV Tasks

| PPL | Qwen0.6 | | Qwen1.7B | | Qwen4B | |
|---|---|---|---|---|---|---|
| | C4 | WiKi | C4 | WiKi | C4 | WiKi |
| GPTQ | 33.12 | 32.12 | 22.19 | 21.05 | 20.06 | 14.53 |
| SmoothQuant | 49.58 | 47.21 | 30.28 | 29.55 | 17.53 | 22.43 |
| K-Quants | 37.87 | 36.53 | 23.44 | 25.32 | 25.47 | 16.78 |
| imatrix | 34.68 | 34.15 | 27.32 | 27.89 | 21.32 | 18.32 |
| MoQE(Our) | 31.98 | 31.67 | 21.72 | 20.44 | 17.09 | 13.89 |
| FP16 | 17.18 | 12.76 | 13.48 | 9.53 | 11.84 | 7.95 |

Table 3: MoQE System with Int4 Quantization Experts

The experimental results are shown in Table 2. As can be observed from the experimental results in Table 2, the MoQE system demonstrates consistently superior performance across various quantization schemes on both ResNet and MobileNet series models. Overall, MoQE effectively mitigates the accuracy degradation typically induced by quantization while maintaining model efficiency.

A detailed analysis reveals the following: On ResNet50, MoQE achieves an accuracy of 78.01%, significantly outperforming BRECQ (76.44%) and DSConv (76.08%), and slightly surpassing N2UQ (76.40%) and QAT (77.09%). This indicates that MoQE more effectively preserves the performance of the original FP16 model (80.21%). The advantage of MoQE becomes even more pronounced on the deeper ResNet101, where it attains an accuracy of 78.91%. This result is notably higher than those of BRECQ(77.60%) and QAT (77.50%), and also exceeds the performance of N2UQ (78.30%) and DSConv (77.90%), fully demonstrating MoQE's robustness and effectiveness in handling complex models. For the lightweight MobileNet model, although the performance gap among different methods is relatively small, MoQE still achieves the best result with an accuracy of 71.36%, outperforming BRECQ (69.10%), QAT (71.10%), N2UQ (71.11%), and DSConv (70.32%), and closely approaching the original FP16 model's accuracy of 71.80%.

In summary, the MoQE system exhibits consistent and superior quantization performance across CV models with varying architectures and complexities. It maximizes the preservation of model accuracy at the Int8 quantization level, outperforming mainstream quantization approaches such as post-training quantization (PTQ), quantization-aware training (QAT), and other specialized methods (e.g., BRECQ, N2UQ and DSConv). These results validate the effectiveness of MoQE as a high-efficiency quantization framework.

### 4.3 CONFIGURATION EXPERIMENT

#### 4.3.1 INT4 EXPERTS

In this section, we will present the impact of various quantization level on the MoQE system. We use Int4 quantization models to build MoQE system. The results are shown in Table 3.

As shown in Table 3, this experiment evaluates the performance of the MoQE system employing various Int4 quantization methods as experts on the Qwen series models (0.6B, 1.7B, 4B). The evaluation metric is perplexity (lower is better), and the results are compared with those of individual quantization methods and the full-precision FP16 model.

The results indicate that the MoQE still exhibits significant advantages at the Int4 quantization level. Across all tested models and datasets, MoQE achieves the lowest perplexity, significantly outperforming each individual quantization method. Specifically, on the Qwen0.6B model, MoQE attains perplexities of 31.98 (C4) and 31.67 (WikiText-2), which are not only lower than those of GPTQ, K-Quants, imatrix, and SmoothQuant, but also closer to the FP16 baseline. This indicates that MoQE can effectively mitigate the severe performance degradation caused by Int4 quantization.

This advantage is equally pronounced in larger-scale models. On the Qwen1.7B, MoQE achieves perplexities of 21.72 (C4) and 20.44 (WikiText-2), surpassing all baseline methods, including the

relatively better-performing imatrix and K-Quants. Notably, on the Qwen4B model with even larger parameter sizes, MoQE's advantage becomes more prominent, with perplexities of 17.09 (C4) and 13.89 (WikiText-2), which not only significantly outperform other quantization methods but also exhibit the smallest gap relative to the FP16 baseline. This demonstrates the MoQE framework's enhanced robustness and higher accuracy retention when handling large models under Int4 quantization.

In summary, the MoQE system successfully overcomes the limitations of individual quantization methods by effectively integrating the predictive capabilities of multiple Int4 quantization experts. It maximizes the preservation of the original model's language understanding and generation abilities while maintaining a high degree of model compression. The consistent superior performance across different scales of Qwen models fully validates the effectiveness and scalability of this framework in extreme quantization (Int4) scenarios.

Furthermore, by comparing Table 1, we observe that after employing Int4 quantization models, the perplexity (ppl) of the individual quantization experts increases, consequently leading to a performance degradation in the MoQE system composed of Int4 experts. However, it is evident that even under these conditions, MoQE still achieves the best performance among all the models evaluated. Notably, MoQE exhibits significantly more pronounced performance gains in the 4-bit quantization setting compared to the 8-bit setting. This is because Int4 quantization exacerbates the bias unfairness among different expert models, thereby providing a larger margin for performance improvement. For a rigorous theoretical derivation regarding this phenomenon and extensive evaluations on MMLU, please refer to Appendix A.4.

### 4.3.2 NUMBER OF EXPERTS

In this section, we investigate the impact of varying the number of experts on the performance of the MoQE system. We further employ the AWQ algorithm to quantize the original full-precision model and incorporate it as a new expert into the MoQE framework. The augmented system is then trained, and the results are presented in the following Table 4.

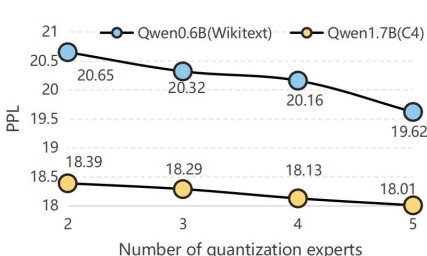

| PPL | Qwen0.6 | | Qwen1.7B | |
|---|---|---|---|---|
| | C4 | Wiki | C4 | Wiki |
| GPTQ | 25.63 | 21.07 | 19.32 | 16.85 |
| SmoothQuant | 25.94 | 22.11 | 18.96 | 16.73 |
| K-Quants | 32.38 | 24.54 | 21.40 | 17.58 |
| imatrix | 27.84 | 26.38 | 19.24 | 18.54 |
| AWQ | 25.54 | 21.04 | 19.23 | 16.78 |
| MoQE (Our) | 24.69 | 19.62 | 18.01 | 15.86 |
| FP16 | 17.18 | 12.76 | 13.48 | 9.53 |

Figure 4: Number of quantization experts          Table 4: MoQE with five quantization experts

As shown in Table 4, we extend the MoQE system by incorporating AWQ as an additional quantization expert, resulting in a total of five experts.

The AWQ method performs competitively among the individual quantization approaches, achieving perplexities of 25.54 (C4) and 21.04 (WikiText-2) on Qwen0.6B, and 19.23 (C4) and 16.78 (WikiText-2) on Qwen1.7B. These results are slightly better than GPTQ and SmoothQuant, indicating AWQ's effectiveness in preserving model accuracy under Int8 quantization.

More importantly, the MoQE system, which integrates routing over all five experts including AWQ, achieves the lowest perplexity across both models and datasets: 24.69 (C4) and 19.62 (WikiText-2) for Qwen0.6B, and 18.01 (C4) and 15.86 (WikiText-2) for Qwen1.7B. This demonstrates that the inclusion of AWQ as an expert further enhances the representational capacity of the ensemble, enabling the router model to leverage complementary strengths across diverse quantization strategies. As shown in Fig 4, increasing the number of experts from 2 to 5 exhibits a trend of performance improvement for the MoQE system, indicating that greater expert diversity yields finer routing and consistently surpasses the best single quantization expert. The superior performance of MoQE confirms that the dynamic expert selection mechanism effectively combines the best aspects

of each method—particularly benefiting from AWQ's strong baseline performance—thereby minimizing quantization error and outperforming even the best individual expert (AWQ) in all cases. This validates the scalability and robustness of the MoQE framework when extended to a larger set of heterogeneous quantization experts.

## 4.4 END-TO-END INFERENCE TIME AND MEMORY CONSUMPTION ANALYSIS

We measured the end-to-end inference latency of MoQE system to determine whether it would result in significant additional overhead. We primarily tested the Extra Inference Time (abbr. EIT), which includes the runtime of the router model and the time required to load the model into VRAM. We expect the unactivated quantized expert models to reside in main memory, while the activated experts are in GPU memory. Therefore, the additional inference time includes the time for loading experts and the time for the router. All inference-latency analyses were performed on a single NVIDIA V100S GPU. As shown in Table 5 that The maximum EIT ratio remains within 8.0%,

| Model | Expert | EIT | EIT Ratio |
|---|---|---|---|
| Qwen 0.6B | 2929.2ms | 27.29ms | 0.93% |
| Qwen 1.7B | 5500ms | 59.0ms | 1.07% |
| LLaMA 3B | 9364.8ms | 222.68ms | 2.38% |
| Qwen 4B | 11606.4ms | 157.4ms | 1.36% |

(a) Extra Inference Time ratio on NLP task.

| Model | Expert | EIT | EIT Ratio |
|---|---|---|---|
| ResNet-50 | 1.64ms | 0.076ms | 4.63% |
| ResNet-101 | 3.90ms | 0.077ms | 1.97% |
| MobileNetV2 | 0.925ms | 0.074ms | 8.0% |

(b) Extra Inference Time ratio on CV task.

Table 5: Inference efficiency analysis of MoQE system.

| Model | Resource Type | MoQE | Single | Total | Increment |
|---|---|---|---|---|---|
| Qwen-1.7B | GPU VRAM | ~7.86 GB | ~7.64 GB | 80G | +0.22 GB (**+0.2%**) |
| | CPU RAM | ~5.17 GB | ~0 GB | 128G~2T | +5.17 GB (+0.2%~4%) |
| LLAMA-3B | GPU VRAM | ~10.43 GB | ~10.17 GB | 80G | +0.26 GB (**+0.3%**) |
| | CPU RAM | ~9.43 GB | ~0 GB | 128G~2T | +9.43 GB (+0.4%~7%) |
| Qwen-4B | GPU VRAM | ~14.76 GB | ~14.30 GB | 80G | +0.46 GB (**+0.5%**) |
| | CPU RAM | ~12.23 GB | ~0 GB | 128G~2T | +12.23 GB (+0.6%~9%) |

Table 6: Resource consumption analysis for the 4-expert MoQE configuration.

while the majority of the EIT ratio is below 3.0%. Such negligible latency confirms that MoQE increases accuracy without sacrificing the real-time responsiveness required for edge scenarios, fully preserving the efficiency advantages of the underlying quantized network. In MoQE system, the GPU keeps only one quantization expert and a lightweight router resident, the remaining experts are not co-resident and are loaded on demand. As shown in Table 6, although MoQE utilizes additional CPU RAM to store inactive experts, this overhead is marginal. Considering that mainstream inference servers typically possess vast CPU RAM resources (often 256GB to 2TB), this increment typically occupies less than 5% of the total system memory. Consequently, utilizing CPU RAM to maintain near-zero VRAM growth represents a highly efficient strategy, enabling MoQE to achieve significant precision improvements without necessitating expensive GPU hardware upgrades. The peak VRAM and CPU RAM of MoQE system closely match those of a single quantization expert during inference. Detailed analyses of VRAM and CPU RAM are provided in the Appendix A.8.

## 5 CONCLUSION

This paper addresses the issue of performance degradation in quantization models by proposing the MoQE system. This framework integrates multiple quantization models and routes input data to the model with the smallest performance degradation, thereby ensuring that the system's overall output surpasses that of any individual quantization expert models. We design distinct router models for CV and NLP tasks. Since inference involves only one quantization model and the router model is relatively small, the overall inference performance is not significantly impacted. Experiments demonstrate that the performance of the MoQE system exceeds that of any single quantization expert within the system.

## 6 REPRODUCIBILITY STATEMENT

We provide an anonymized, downloadable code repository in https://github.com/paper-submission-goii/MoQE that includes complete scripts and instructions for constructing/loading quantization experts and for training/evaluation. The repository also contains environment files (exact dependency versions), data-preprocessing steps, and the hardware specifications used in our experiments, enabling reproduction of the main experimental results and ablations. All datasets are publicly available and used under their original licenses. Experiments code address is https://github.com/paper-submission-goii/MoQE.

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

# A APPENDIX

## A.1 EXPERIMENTS ROUTER MODEL TRAINING DETAIL

### A.1.1 CV ROUTER MODEL TRAINING DETAIL

In our CV router model training process, the primary objective is to end-to-end train a router model to learn how to intelligently and dynamically assign weights to a fixed set of experts based on the input data. To achieve fine-grained optimization of the router model, we employ the AdamW optimizer. We implement a staged dynamic learning rate strategy: during the initial 1 to 5 training epochs, the learning rate for the router model is set to 1e-5; from epochs 6 to 10, it is increased to 2e-5 to accelerate convergence; thereafter, it reverts to 1e-5 for fine-tuning. Additionally, an early stopping mechanism is implemented: if the validation accuracy does not improve for seven consecutive epochs, training will automatically terminate. This is complemented by a strategy to dynamically reduce the learning rate when accuracy plateaus, ensuring the model ultimately converges to its optimal performance.

### A.1.2 NLP ROUTER MODEL TRAINING DETAIL

In the training of the router model, the model weights are initialized using a normal distribution, and the learning rate is set to $5 \times 10^{-5}$. We employ the 8-bit AdamW optimizer, which has a lower memory footprint, combined with a cosine annealing learning rate scheduler featuring a 2000-step warm-up phase. The total loss function consists of two components: the standard cross-entropy loss and an adaptive load balancing loss. The weighting coefficient $\alpha$ for this auxiliary loss term is dynamically adjusted based on the degree of balance in expert activation during training (with a base value of 0.02); when imbalances in expert utilization occur, the penalty strength of this term increases accordingly, thereby encouraging the model to utilize all experts more uniformly. Regarding the training procedure, the model is trained for a total of 10 epochs. A curriculum learning strategy is adopted: during the first two epochs, the model is trained only on a subset of 70,000 samples; starting from the third epoch, it transitions to training on the full dataset, which is more challenging. We achieve an effective batch size of 48 by setting the per-device batch size to 8 and performing gradient accumulation over 6 steps, while applying gradient clipping with a maximum norm of 1.0 to ensure training stability.

## A.2 THE BIASES EXHIBITED BY DIFFERENT QUANTIZATION MODELS ON CV TASKS

This section supplements the observation that different Int8 quantization experts exhibit distinct biases in the data space for CV tasks. To validate this hypothesis, we constructed seven mutually exclusive sub-datasets from the ImageNet-1K dataset (each containing 5k–7k images with balanced label distributions). On this basis, we applied four mainstream Int8 quantization methods—BRECQ, QAT, N2UQ, and DSConv—to the ResNet-50 FP32 weights, ensuring that the data, hyper-parameters, and inference pipeline were completely identical, thereby excluding external variables from interfering with performance differences.

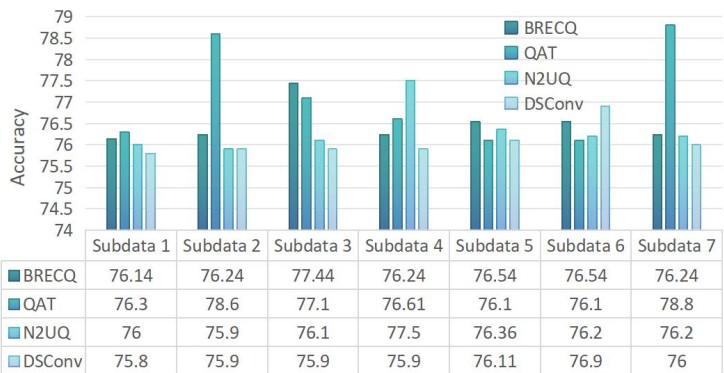

| | Subdata 1 | Subdata 2 | Subdata 3 | Subdata 4 | Subdata 5 | Subdata 6 | Subdata 7 |
|---|---|---|---|---|---|---|---|
| BRECQ | 76.14 | 76.24 | 77.44 | 76.24 | 76.54 | 76.54 | 76.24 |
| QAT | 76.3 | 78.6 | 77.1 | 76.61 | 76.1 | 76.1 | 78.8 |
| N2UQ | 76 | 75.9 | 76.1 | 77.5 | 76.36 | 76.2 | 76.2 |
| DSConv | 75.8 | 75.9 | 75.9 | 75.9 | 76.11 | 76.9 | 76 |

Figure 5: Top-1 accuracy of four Int8 quantization methods on seven ImageNet-1K sub-datasets.

The results are shown in the figure 5. By comparing the Top-1 accuracy of different quantization experts on each sub-dataset, we can see that for the same subset the performance of different quantization methods differs, and the relative performance ranking of each quantization model changes across different subsets; no single quantization method remains optimal on all subsets.For instance, on Subdataset3, N2UQ outperforms BRECQ by approximately 2.5% in Top-1 accuracy, whereas on Subdataset5, BRECQ surpasses N2UQ by 1.8%. The maximum gap across subsets reaches 4.7%, highlighting the strong data-dependent nature of quantization. This result empirically corroborates the core viewpoint proposed in Section 3.1 of the main text: that different quantization experts display bias distributions across the data, which provides the feasibility for dynamically assigning samples to the most suitable expert in the MoQE system.

## A.3 THEORETICAL PROOF OF MoQE ERROR BOUND

### DEFINITION OF LOSS UPPER BOUND

We define the loss upper bound function $L_k(a)$ Liu et al. (2025) for the $k$-th quantization expert on a specific data subset $a \in \mathcal{A}$ as:

$$L_k(a) \triangleq \frac{1}{2}\sqrt{n}(s_{max}^{(k)})^2 \cdot \|g_{w^*}^{D_a}\| + \frac{1}{8}n(s_{max}^{(k)})^4 \cdot \mathrm{Tr}(H_{w^*}^{D_a}) + \mathcal{O}(\|\Delta w^*\|^3) \tag{5}$$

where

- $w^*$: the well-trained model parameters;
- $g_{w^*}^{D_a} = \nabla_w L(w^*; D_a)$: the gradient of loss function $L$ at $w^*$ on subset $D_a$;
- $\mathrm{Tr}(H_{w^*}^{D_a})$: the trace of the Hessian matrix of function $L$ at $w^*$ on subset $D_a$;
- $\Delta w^*$: the quantization error;
- $s_{max}$: maximum noise by quantization;
- $n$: the dimension of the vector $w^*$;

$\mathcal{E} = \{E_1, ..., E_K\}$: the set of $K$ quantization experts. A single quantization model can be viewed as the system using the same fixed quantization expert $E_{fixed}$ (where $fixed \in \{1, ..., K\}$) on all data.

The total loss $\mathbb{L}_{Single}$ is the sum of losses for all subsets:

$$\mathbb{L}_{Single} = \sum_{a \in \mathcal{A}} L_{fixed}(a) \tag{6}$$

The MoQE system employs a routing strategy to select the optimal expert $k^*(a)$ from the quantization expert pool for each data subset $a$:

$$k^*(a) = \arg \min_{k \in \{1,...,K\}} L_k(a) \tag{7}$$

The total loss of the MoQE system is:

$$\mathbb{L}_{MoQE} = \sum_{a \in \mathcal{A}} L_{k^*(a)}(a) = \sum_{a \in \mathcal{A}} \left( \min_{k \in \{1,...,K\}} L_k(a) \right) \tag{8}$$

The MoQE Router iterates through all experts and selects the one with the smallest loss. Therefore, for any subset $a$:

$$\min_{k \in \{1,...,K\}} L_k(a) \leq L_{fixed}(a) \tag{9}$$

Summing Equation (9) over all subsets $a \in \mathcal{A}$ gives:

$$\sum_{a \in \mathcal{A}} \left( \min_{k \in \{1,...,K\}} L_k(a) \right) \leq \sum_{a \in \mathcal{A}} L_{fixed}(a) \tag{10}$$

Substituting Equation (6) and Equation (8), we obtain:

$$\mathbb{L}_{MoQE} \leq \mathbb{L}_{Single} \tag{11}$$

This guarantees that the total loss upper bound of the MoQE system is always less than or equal to that of any single quantation model.

MoQE will fail if for every subset $a \in \mathcal{A}$:

$$\arg \min_{k \in \{1,...,K\}} L_k(a) = fixed \tag{12}$$

This means that the fixed expert $E_{fixed}$ coincidentally is the best performer among all quantization experts for every subset. However, the variation in the loss function for a quantization model on a specific data subset is determined by $\|g_{w^*}^{D_a}\|$ and $\text{Tr}(H_{w^*}^{D_a})$. Since the real data distribution is complex, different subsets $D_a$ exhibit radically different $\|g_{w^*}^{D_a}\|$ and $\text{Tr}(H_{w^*}^{D_a})$, thus this equality case will not hold true in real-world scenarios.

## A.4 ADDITIONAL EVALUATION ON MMLU

To comprehensively evaluate the performance improvement of MoQE in downstream tasks, we extended the evaluation scope to the MMLU dataset, covering both 4-bit and 8-bit settings. As shown in Table 7, MoQE (4-bit) consistently exceeds all quantization expert baselines with a significant advantage, while MoQE (8-bit), although the magnitude of improvement is not large, further narrows the score gap with the upper limit of FP16.

| Method | 4-bit Quantization | | | | 8-bit Quantization | |
|---|---|---|---|---|---|---|
| | Qwen-0.6B | Qwen-1.7B | Qwen-4B | Qwen-14B | Qwen-4B | Llama-3B |
| FP16 (Bound) | 47.1 | 60.0 | 69.7 | 78.5 | 69.7 | 63.4 |
| AWQ | 43.1 | 53.9 | 66.0 | 76.3 | 69.5 | 63.1 |
| GPTQ | 40.0 | 52.8 | 65.8 | 75.9 | 59.9 | 61.7 |
| K-Quants | 32.2 | 41.8 | 62.6 | 68.4 | 64.2 | 60.4 |
| SmoothQuant | 30.8 | 44.1 | 59.2 | 71.3 | 69.3 | 62.3 |
| imatrix | 38.4 | 47.5 | 65.1 | 70.6 | 69.4 | 61.4 |
| **MoQE (Ours)** | **45.5** | **55.9** | **68.3** | **77.9** | **69.7** | **63.3** |

Table 7: MMLU scores under 4-bit and 8-bit quantization.

Based on the analysis of Eq. 5, the quantization loss is determined by three terms: $s_{\max}$, $g_{w^*}^{D_a}$, and $H_{w^*}^{D_a}$. Compared with Int8, Int4 quantization leads to a significantly larger $s_{\max}$. Under identical dataset conditions, this amplified noise significantly exacerbates the bias unfairness among different quantization experts. Consequently, MoQE demonstrates superior efficacy specifically in low-precision regimes.

### A.5 ROUTING PROBABILITY ASSIGNED TO EACH QUANTIZATION EXPERT

To better understand how the router model utilizes different quantization experts, we study the routing probabilities assigned during inference. For each sample, the router model will output a probability distribution about the quantization experts, which reflects the suitability of each quantization expert for the assigned sample. Despite differences between NLP and CV, both tasks demonstrate adaptive routing behaviors rather than static or random allocations. Figures 6- 7 provide clear evidence of this phenomenon.

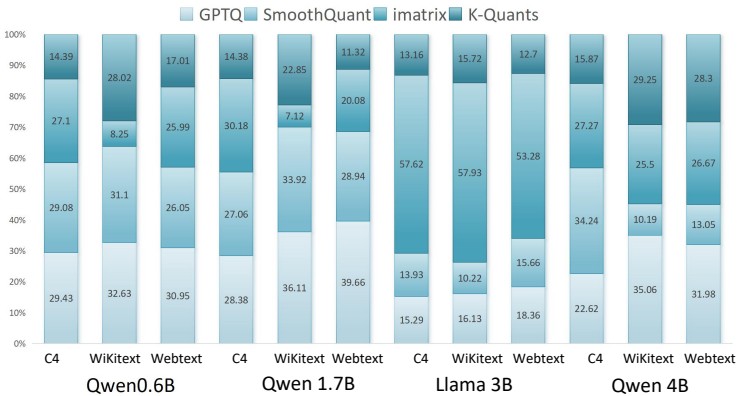

Figure 6: Routing probability distributions for NLP tasks.

In NLP tasks (Figure 6), the router model displays strong sensitivity to input semantics. For certain datasets, the routing distribution is more concentrated, indicating that the router model consistently identifies one quantization expert as more reliable for specific linguistic patterns. In contrast, for other datasets, the probability mass is spread more evenly, suggesting that multiple quantization experts provide complementary strengths. These various probability distributions indicate that the router model can adaptively select the most suitable quantization expert based on the semantic context.

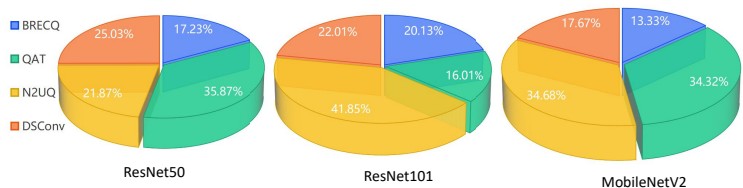

Figure 7: Routing probability distributions for CV tasks.

In CV tasks (Figures 7), for deeper architectures such as ResNet101, routing probabilities reveal clearer specialization: inputs are more consistently assigned to a subset of experts, which aligns with the stronger bias of quantization methods on complex visual features. Conversely, for lightweight architectures such as MobileNetV2, the distributions are smoother and more balanced, implying that experts perform more comparably when the model has less representational complexity.

These results are consistent with the analysis in Appendix A.2, which demonstrated that different quantization models exhibit data-dependent biases. The routing probabilities show that the router model effectively captures these biases and adaptively allocates inputs to the most suitable expert.

Overall, this probabilistic mechanism enables MoQE to integrate heterogeneous quantization methods and consistently achieve better performance than any single expert. These routing distributions further confirm that the proposed load-balancing design successfully prevents expert collapse during training, ensuring stable and diverse utilization of all experts.

## A.6   THE MoQE SYSTEM WITH DIFFERENT NUMBERS OF QUANTIZATION EXPERTS

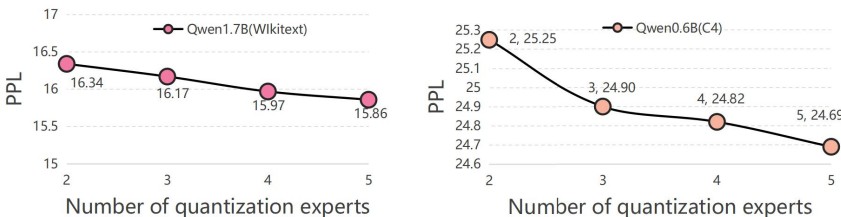

Figure 8: Impact of quantization expert size on MoQE system.

This section provides additional analysis of MoQE under different numbers of quantization experts (2–5). As illustrated in Figures 8 , the results show a clear trend: increasing the number of experts from two to five brings improvements for both Qwen-0.6B and Qwen-1.7B, as the router model can utilize a richer library of quantization experts to make more accurate allocations. When adding AWQ as the fifth expert, MoQE system continues to improve. The MoQE system with five experts achieves the best results, surpassing the strongest single quantization expert, which confirms that MoQE system can effectively integrate the complementary advantages of different quantzation experts. Importantly, since each input only activates one quantization expert, the increase in the number of experts does not introduce significant inference overhead, which has been confirmed in the latency analysis. In summary, these data indicate that the performance of MoQE system steadily improves as the number of experts increases.

## A.7   CONVERGENCE OBSERVATIONS AND ANALYSIS OF THE MoQE ROUTER MODEL

This section tracks the evolution of routing accuracy and Top-1 accuracy over the training process to analyze whether the router model stably assigns samples to the most suitable quantization expert, thereby validating the overall effectiveness of MoQE on CV tasks. In our setup, the expert weights remain frozen, and only the router model is fine-tuned, so as to avoid the confounding effect of "expert retraining" on the routing evaluation. We compute the corresponding metrics on the validation set throughout training and plot the curves to observe the convergence and robustness of the routing process.

Let the validation set be $\mathcal{D} = \{(x_i, y_i)\}_{i=1}^N$, and let $\{f_j\}_{j=1}^M$ denote $M$ frozen quantization experts. For a sample $(x_i, y_i)$, define the per-sample evaluation loss of expert $f_j$ as $\ell_j(x_i, y_i)$ (aligned with the task metric; lower is better). The optimal expert is

$$j^\star(x_i) = \arg\min_{j \in [M]} \ell_j(x_i, y_i), \qquad [M] := \{1, \ldots, M\}.$$

Given a router model $r : \mathcal{X} \to [M]$, the *Routing Accuracy* is

$$\text{RA} = \frac{1}{N} \sum_{i=1}^N \mathbf{1}\{\, r(x_i) = j^\star(x_i) \,\}.$$

**Indicator function.**   We denote by $\mathbf{1}\{\cdot\}$ the indicator function:

$$\mathbf{1}\{A\} = \begin{cases} 1, & \text{if the statement } A \text{ is true,} \\ 0, & \text{otherwise.} \end{cases}$$

Therefore, $\mathbf{1}\{\, r(x_i) = j^\star(x_i) \,\} = 1$ if the router model picks the same expert as the optimal expert for sample $x_i$, and 0 otherwise.

To evaluate the router model's learning process, we perform periodic evaluations on a fixed validation set throughout training. The system contains $N$ pre-quantized and frozen experts. We first annotate the optimal expert for each validation sample: for each sample $(x_i, y_i)$, we run a forward pass with all $N$ experts, compute the task-aligned evaluation loss, and record the expert with the smallest loss as that sample's $j^*(x_i)$. We then monitor two key metrics: (a) Routing Accuracy (RA), the proportion of samples for which the router's choice $r(x_i)$ matches the annotated optimal expert $j^*(x_i)$; and (b) Top-1 accuracy, an end-to-end metric obtained by using the router selected expert $f_{r(x_i)}$ to produce the prediction and comparing it with the ground-truth label $y_i$. We log both metrics at fixed intervals during training, thereby depicting the router's approximation and convergence toward the "optimal expert" decision and its resulting impact on the system's overall Top-1 accuracy.

| Model | System Convergence Upper Bound | MoQE system | $\Delta$ to Upper Bound |
|---|---|---|---|
| ResNet-50 | $\approx 78.44\%$ | 78.01% | 0.43% |
| ResNet-101 | $\approx 79.23\%$ | 78.91% | 0.32% |
| MobileNetV2 | $\approx 71.51\%$ | 71.36% | 0.15% |

Table 8: System convergence upper bound

Table 8 presents the system convergence upper bound measured by selecting the optimal expert per sample on the validation set. As shown, our router performs very well: across multiple models, the gap to this upper bound is small, and on MobileNetV2 the gap is only 0.15%.

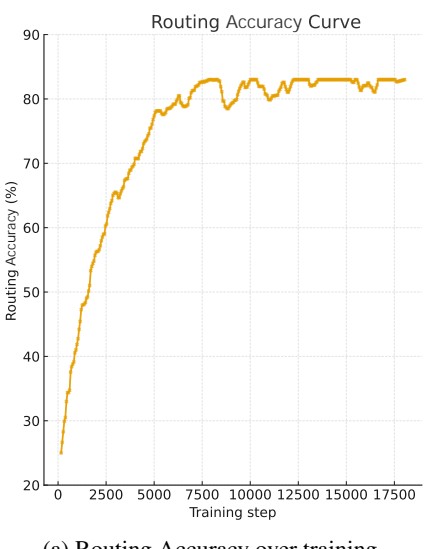

(a) Routing Accuracy over training

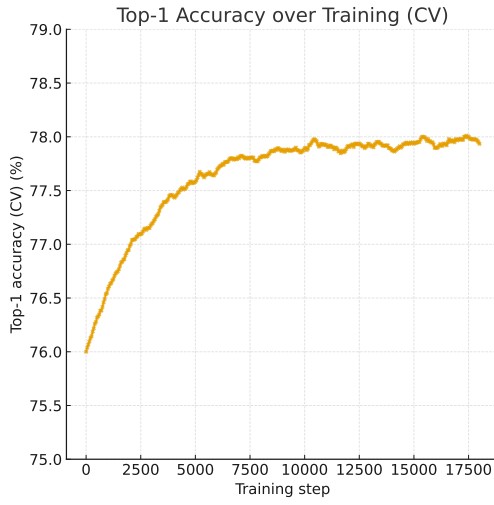

(b) Top-1 accuracy over training (CV)

Figure 9: Routing convergence trends on CV tasks (a) Router Accuracy (RA) during training in the MoQE system. (b) Top-1 accuracy during training in the MoQE system

As shown in Figure 9 (a), the curve exhibits a clear three-phase pattern: in the early stage of training, RA (routing accuracy) is about 25%; once training begins, it rapidly jumps to 60%. In the mid stage, it continues to rise steadily with small, controllable fluctuations (attributable to the trade-off between the load-balancing strategy). Finally, RA (routing accuracy) stably converges around 83%. Throughout this process, the quantization experts derived from the same base model are frozen and only the router is fine-tuned, which also demonstrates the effectiveness of the MoQE system. The improvement in RA (routing accuracy) can be directly interpreted as the router more frequently selecting the most suitable quantization expert for the task at hand, indicating that specialization matching to different sub-distributions is progressively taking shape.

As shown in Figure 9 (b), The MoQE system's Top-1 accuracy increases during training (from approximately 76% to 78%), with the rise of router accuracy in Figure 9 (a): the router more often assigns samples to the quantization expert that is better at the corresponding subdistribution, thus making the overall output distribution more aligned with the task objective and yielding a stable gain in Top-1 accuracy. This indicates that MoQE system's gains indeed come from better routing rather than "stacking expert capacity".

Figure 9 directly support the core proposition of the MoQE system: dynamic routing can enhance the accuracy of the quantization model. As training progresses, the probability that the router assigns samples to the most suitable quantization expert steadily rises to around 83%, and this is accompanied by a concurrent increase in the MoQE system's Top-1 accuracy. The experiments in this section indicate that the Int8 expert models obtained through different quantization strategies based on the same base model show biases in the data distribution . The router learns from this and adaptively allocates the samples to the most suitable quantization experts.

## A.8 PEAK VRAM AND CPU RAM USAGE ANALYSIS

This section focuses on runtime memory (the peak usage of CPU RAM and VRAM during inference or training). These metrics directly impact deployment feasibility and real-time performance (latency and throughput). Accordingly, we report and analyze peak VRAM and runtime CPU RAM to assess the practical deployment efficiency of MoQE system.

This subsection does not focus on offline data storage or disk space usage. Disk space usage has no direct impact on runtime memory or real-time performance and can be evaluated independently. Under MoQE system's on-demand loading mechanism, the GPU keeps only a lightweight router and one Int8 expert resident , the remaining quantization experts are not resident in VRAM and are loaded only when an expert switch occurs. Meanwhile,disk storage has good scalability and large capacity—its capacity can even be measured in TB , which is significantly higher than the order of magnitude of VRAM and CPU RAM. We can increase disk capacity by upgrading local storage, mounting multiple disks, or using network-based storage solutions. Therefore, in common deployments, the size of offline weights does not constrain runtime memory or real-time performance , and this is not a significant obstacle for the MoQE system. Therefore, we consider the MoQE system's greater disk space consumption to be an acceptable form of resource consumption.

As shown in Table 9,we report the storage size of model weights for different models under FP16/Int8/Int4. The router model's weight size is very small: 16.7 MB (full precision) for the NLP router and 2.0 MB (full precision) for the CV router. Relative to the corresponding Int8 quantization experts, the router typically accounts for less than a few percent on large models for example, Qwen-1.7B: 1.70 GB ( 1% ), LLaMA-3B: 3.00 GB ( 0.6% ), Qwen-4B: 4.00 GB ( 0.4% ). On the CV tasks, ResNet-50: 25.6 MB ( 7.8% ) and ResNet-101: 44.5 MB ( 4.5% ). For very small backbones (e.g., MobileNetV2: 14.2 MB), the relative share increases, but the absolute router size still remains about 2 MB. Moreover, peak VRAM during inference is dominated by activations, so the router model's extra cost in memory and latency is negligible. Combined with the MoQE system inference strategy (the GPU keeps only the lightweight router and the currently selected single Int8 quantization expert resident), MoQE system is on par with single expert Int8 in both storage and runtime memory, while achieving stable accuracy gains through better dynamic routing.

| Category | Model | FP16 | Int8 | Int4 |
|---|---|---|---|---|
| NLP | Qwen-0.6B | 1.20 GB | 0.60 GB | 0.30 GB |
| | Qwen-1.7B | 3.40 GB | 1.70 GB | 0.85 GB |
| | LLaMA-3B | 6.00 GB | 3.00 GB | 1.50 GB |
| | Qwen-4B | 8.00 GB | 4.00 GB | 2.00 GB |
| CV | ResNet-50 | 51.2 MB | 25.6 MB | 12.8 MB |
| | ResNet-101 | 89.0 MB | 44.5 MB | 22.3 MB |
| | MobileNetV2 | 14.2 MB | 3.4 MB | 1.7 MB |
| Router | NLP Router (full precision) | 16.7 MB | – | – |
| | CV Router (full precision) | 2.0 MB | – | – |

Table 9: Memory required to store the weights of different models.

| Model | VRAM (GB) | | | CPU RAM | |
|---|---|---|---|---|---|
| | MoQE (Int8) | Single (Int8) | Single (FP16) | MoQE | Single (Int8/FP16) |
| ResNet-50 | ≈ 0.94 | ≈ 0.90 | ≈ 0.93 | ≈ 76.8 MB | ≈ 0 (slight) |
| ResNet-101 | ≈ 1.35 | ≈ 1.30 | ≈ 1.34 | ≈ 133.5 MB | ≈ 0 (slight) |
| MobileNetV2 | ≈ 0.38 | ≈ 0.35 | ≈ 0.36 | ≈ 10.2 MB | ≈ 0 (slight) |
| Qwen-0.6B | ≈ 4.32 | ≈ 3.96 | ≈ 4.62 | ≈ 1.84 GB | ≈ 0 (slight) |
| Qwen-1.7B | ≈ 7.86 | ≈ 7.64 | ≈ 9.43 | ≈ 5.17 GB | ≈ 0 (slight) |
| LLaMA-3B | ≈ 10.43 | ≈ 10.17 | ≈ 13.47 | ≈ 9.43 GB | ≈ 0 (slight) |
| Qwen-4B | ≈ 14.76 | ≈ 14.30 | ≈ 18.53 | ≈ 12.23 GB | ≈ 0 (slight) |

Table 10: Peak VRAM and CPU RAM usage during inference for both CV and NLP tasks.

Compared to a single quantization model, the incremental cost of MoQE system is concentrated in the CPU RAM and disk space . As shown in Table 10, the additional CPU RAM in NLP tasks is Qwen-0.6B: 1.84 GB, Qwen-1.7B: 5.17 GB, LLaMA-3B: 9.43 GB, Qwen-4B: 12.23 GB , in CV tasks, it is ResNet-50: 76.8 MB, ResNet-101: 133.5 MB, MobileNetV2: 10.2 MB. These values scale linearly with the size of the expert and occur only in CPU RAM and disk space. The aspect of disk space usage was previously analyzed.

The MoQE peak VRAM is effectively identical to the single Int8 baseline: for NLP, the difference is approximately 0.2 to 0.5GB, and for CV it is no more than 50MB, indicating no additional VRAM pressure. Relative to the FP16 baseline (Table 10), peak VRAM in NLP is substantially higher than with MoQE system as much as about 3.8 GB highlighting that, under the same batch size, peak VRAM is dominated by KV cache and intermediate activations rather than by weights. Under the same memory and latency budget, MoQE matches the single Int8 quantization model on runtime footprint and latency while surpassing it in Top-1 accuracy through higher routing accuracy that assigns each sample to the most suitable quantization expert. Relative to the FP16 baseline, MoQE system in can also reduce peak VRAM.

| Model | Expert (ms) | Router (ms) | (I/O) (ms) | I/O% | Total% (Router+I/O) |
|---|---|---|---|---|---|
| Qwen-0.6B | 2929.2 | 26.77 | 41.07 | 1.40% | 2.32% |
| Qwen-1.7B | 5500.0 | 57.60 | 121.45 | 2.21% | 3.26% |
| LLaMA-3B | 9364.8 | 220.05 | 212.78 | 2.27% | 4.62% |
| Qwen-4B | 11606.4 | 154.20 | 312.54 | 2.69% | 4.02% |

Table 11: Router model and expert loading overhead ratios.

The MoQE system keeps one quantization expert resident in VRAM, with the remaining quantization experts resident in CPU RAM. The router model makes decisions on the GPU, if the selected quantization expert does not change, no weight switch is performed. Only when a switch is needed are the target quantization expert's weights loaded from CPU RAM to VRAM (detailed data are provided in Table 11). Under this deployment strategy, the router model's decision latency accounts for about 0.9%–2.4% of the quantization expert's runtime, and the quantization expert loading latency (I/O) accounts for about 2%. Because a considerable portion of tasks do not need to switch quantization experts, the actual average cost of expert switching is lower. Even under a worst case switching scenario—where every request switches to a different quantization expert the end-to-end overhead remains within 5%, which we consider acceptable.

In summary , memory is not a bottleneck but an acceptable resource investment. MoQE system achieves no increase in VRAM and negligible speed impact, requiring only controlled CPU RAM (and corresponding disk space usage) to obtain stable accuracy gains.

## A.9 RELEVANT DATASET, MODELS

To ensure the fairness and reproducibility of the experiment, a strict text purification process is carried out during the preprocessing stage: empty lines, HTML tags, URLs, and redundant lines containing repeated characters are removed, and all sequences are uniformly truncated to 2,048 tokens;

the random seed is fixed throughout the process. Five mainstream quantization methods, namely AWQ, SmoothQuant, GPTQ, k-quants, and imatrix, are used in the experiment to quantize the four large language models, Qwen-0.6 B, Qwen-1.7 B, Llama-3 B, and Qwen-4 B, into quantization experts. The relevant weights have been fully open-sourced. The training and evaluation data are taken from three public datasets, C4, WikiText-2, and OpenWebText. All resources and processing scripts have been sourced from the official repository.

### A.10    USE OF LLMS

We used GPT-5 solely for language editing and refinement. All scientific ideas, experiments, and conclusions were conceived and verified by the authors, who take full responsibility for the content.

