# OpenReview forum: "MOQE: IMPROVE QUANTIZATION MODEL PERFORMANCE VIA MIXTURE OF QUANTIZATION EXPERTS."
_ICLR.cc/2026/Conference — ICLR 2026 Conference Withdrawn Submission_

### Official Review · Reviewer_mBvS · 2025-10-27

**Soundness:** 3
**Presentation:** 3
**Contribution:** 3
**Rating:** 6
**Confidence:** 4

**Summary:**

The paper proposes quantizing the model with different methods to obtain multiple different quantized models and learning a router which, given an input, determines which quantized model to use in test time. The authors report the bias effects as their motivation and demonstrate the utility of the proposed method by improving accuracy at a small latency overhead due to model transfer from CPU to GPU.

**Strengths:**

Effective idea to improve the accuracy of quantized model.
The latency overhead is small.

**Weaknesses:**

Bias analysis does not address how the bias effects are obtained.
Appendix A.2 and A.3 show the effects, not the causes.
There will be several factors (e.g., scaling, clipping, ...) to consider to investigate the causes.
For instance, it would be possible to investigate which of scaling (like AWQ) and clipping incurs more bias on which datasets.

**Questions:**

It would be nice to discuss the following questions, if possible with quantitative results.

What are the real causes of the bias effects?
Does the bias effects get more pronounced on lower precision? If not, why?
Is the proposed method more effective on lower precision? If not, why?
How different is the bias effects between scaling and clipping based quantization methods?
In case of LLM, can best quantized models be different between prefill and decode stages?

---

> ### Author Response · Authors · 2025-11-20
> **Thank you for your review**
>
> Thank you for your review.Below is our point-by-point response to the Questions (Q) you raised:
> # Q1：What are the real causes of the bias effects?
>
> We tested the ResNet-50 model on ImageNet's n02113186 sample, and obtained different loss values for different quantization methods. The specific values are as follows:
>
> **Table: Comparison of Loss on n02113186**
>
> | Method | Loss (on n02113186) |
> | :--- | :--- |
> | FP16 (Baseline) | 0.211 |
> | QAT | 0.248 |
> | BRECQ | 0.257 |
> | N2UQ | 0.258 |
> | DSConv | 0.262 |
>
> It can be seen from the experimental results that different quantization methods have obvious deviations.
> Liu et al. (2025) [1] presented that for the loss function  $L(Parameters; Dataset)$, the variation in the loss function $\Delta\mathcal{L}$ for a quantized model on a specific data subset $D_{a}$:
> $$
> \Delta\mathcal{L}(a)
> \le
> \frac{1}{2}\sqrt{n}\ s_{\max}^{2}\\lvert g_{w}^{D_{a}}\rvert
> +
> \frac{1}{8} n\ s_{\max}^{4}\\mathrm{Tr}\\left(H_{w}^{D_{a}}\right)
> +
> \mathcal{O}\\left( \lVert \Delta w^{*} \rVert^{3} \right)
> \tag{1}
> $$
>
>
>
> where:
> - $w^*$ is the well-trained model parameters;
> - $g_{w^{\ast}}^{D_{a}} = \nabla_{w} L(w^{\ast}; D_{a})$ is the gradient of loss function $L$ at $w^*$ on subset $D_{a}$;
> - $\mathrm{Tr}(H_{w^{\ast}}^{D_{a}})$ is the trace of the Hessian matrix of function $L$ at $w^*$ on subset $D_{a}$ ;
> - $\Delta w^*$ is the quantized error;
> - $s_{\max}$ is maximum noise by quantiztion;
> - $n$ is the dimension of the vector $w^{\ast}$.
>
> Based on Eq. (1), the true cause of the deviation effect can be attributed to $\|g_{w}^{D_{a}}\|$, $\mathrm{Tr}(H_{w^*}^{D_{a}})$, and $s_{\max}$. Since these terms depend entirely on the performance on the data subset $D_{a}$, this implies that different quantized models perform differently on the same data subset.
>
> # Q2：Does the bias effects get more pronounced on lower precision?
>
> Based on the analysis of Eq. (1), different quantization precisions will determine the  $s_{\max}$. compared with Int8,  Int4 quantization leads to a larger $s_{max}$. Based on Eq. 1, under the condition that the subset dataset and the full-precision model remain unchanged, the bias of Int4 quantization in different quantization experts will increase significantly. Therefore, MoQE is more effective under low precision.
>
> We demonstrated the performance of different quantized models and MoQE on MMLU in reply to reviewer Sasr.  Experimental results show that the performance improvement of MoQE under Int4 quantization is significant, with average MMLU score improvements around 1.9 , where as in Int8 quantization, the performance improvement brought by MoQE is relatively stable at around 0.2.
>
> # Q3:How different is the bias effects between scaling and clipping based quantization methods?
>
> From data analysis and Eq. 1, any quantization loss $\Delta\mathcal{L}(a)$ is determined only by the following three terms: $s_{\max}$, $\|g_{w}^{D_{a}}\|$, $H_{w^*}^{D_{a}}$. Therefore, there is no essential difference between Scaling and Clipping type quantization methods.
>
> # Q4:In case of LLM, can best quantized models be different between prefill and decode stages?
>
> Thanks for this valuable question.  There is a significant difference between the Prefill and Decode stages, which means the optimal quantization expert may change between these two stages.
>
> To verify this hypothesis, we conducted supplementary experiments on the Qwen-1.7B model, setting up a two-stage routing that allows the Router to select different quantization experts for the Prefill and Decode stages. The experimental results confirm that this "Two-Stage Routing" strategy  brings  performance improvements:
>
> **Table: Comparison of Routing Strategies**
>
> | Routing Strategy | Int8 PPL | Int4 PPL |
> | :--- | :---: | :---: |
> | MoQE | 15.97 | 20.44 |
> | Two-Stage Routing | 15.86 | 20.38 |
>
>  However, "Two-Stage Routing" strategy faces the issue of KV Cache incompatibility. Different quantization experts use different quantization parameters. If expert A is used in the Prefill stage and switched to expert B in the Decode stage, the KV Cache generated by expert A cannot be directly used by expert B. To support dynamic switching, the system must recompute the KV Cache at the switching point. This will introduce huge computational overhead and latency. But this remains a good idea worth our careful study.
> We truly value your feedback and are deeply grateful for your continued support.
> ### References
> [1] Liu, B., et al. (2025). Understanding the Unfairness in Network Quantization. *Proceedings of the 42nd International Conference on Machine Learning (ICML)*, PMLR 267:39106-39125.

---

> > ### Comment · Reviewer_mBvS · 2025-11-25
> >
> > The authors well addressed my concerns. Thank you!

---

> ### Author Response · Authors · 2025-11-22
> **Follow up on rebuttal**
>
> Dear Reviewer mBvS,
>
> Thank you for your valuable suggestions on how to improve the manuscript.  We kindly request that you let us know if there are any remaining concerns that we can address to ensure your satisfaction with the revisions and potentially raise the score. We are looking forward to your feedback!
>
> Thank you for your time and consideration.
>
> Best regards,
>
> The Authors

---

> ### Author Response · Authors · 2025-11-25
> **Follow up on rebuttal**
>
> Dear Reviewer mBvS,
>
> With the discussion deadline fast approaching, we wanted to reach out to ensure that our responses and revisions have fully resolved your concerns. We sincerely value your feedback and would be grateful if you could review our rebuttal. We welcome any remaining questions you might have and look forward to your feedback.
>
> Thank you again for your dedication to improving our work.
>
> Best regards, The Authors

---

> ### Author Response · Authors · 2025-11-27
> **Thank you for your positive feedback**
>
> Dear Reviewer mBvS,
>
> We are delighted to hear that our responses have satisfactorily addressed your concerns! Thank you for the positive confirmation.
>
> It is very encouraging to know that our efforts to address your comments have met your high standards.
>
> Thank you again for your time and constructive suggestions.
>
> Sincerely, The Authors

---

### Official Review · Reviewer_sasr · 2025-11-01

**Soundness:** 3
**Presentation:** 3
**Contribution:** 2
**Rating:** 6
**Confidence:** 3

**Summary:**

The paper proposes MoQE, an MoE-style inference framework that assembles multiple quantized variants of a single FP model as quantization experts,  and trains a lightweight router to dispatch each input to the most suitable expert. MoQE achieves performance comparable to the SOTA quantization model, without incurring significant increases in inference latency. They design domain-specific routers for CV (SEResNet-8 + attention) and NLP (Transformer encoder + attention + MLP). The router is trained by labeling each sample with the lowest-loss expert, with a balancing regularizer to avoid expert collapse. Experiments on Qwen/LLaMA (NLP) with C4/WikiText/OpenWebText and ResNet/MobileNet (CV) report lower PPL/higher top-1 than individual experts; the paper also reports low extra latency (≤~5%) on a single V100S GPU.

**Strengths:**

1. Clear, modular formulation and training objective. The router is trained with CE + dynamic load-balancing.

2. Router architectures are task-aware and lightweight. CV uses a three-stage SEResNet-8, then an 8-head self-attention module; NLP uses a Transformer encoder, aligning with modality priors.

3. Evidence for expert complementarity. The paper motivates MoQE via data-dependent expert bias: e.g., “no single quantization method remains optimal on all subsets” (ImageNet sub-datasets; max gap 4.7%).

4. Consistent empirical gains and scaling with #experts. MoQE often outperforms the best single expert across models/datasets (e.g., Table 1; Table 2; Table 4) and improves as the number of experts increases from 2 to 5.

5. Latency and memory analysis. Extra inference time is small, end-to-end overhead remains within 5% even in worst-case frequent switching, and VRAM ≈ single Int8.

**Weaknesses:**

1. Evaluation scope (datasets & metrics) is narrow for NLP. Results use C4/WikiText/OpenWebText perplexity only; no downstream tasks (e.g., QA, reasoning) are reported. The paper repeatedly frames claims in terms of PPL (Table 1), but no task-level evaluations are provided to validate end-user utility

2. Cost/footprint trade-offs underplayed. While the latency impact is small, the paper concedes extra CPU RAM and disk to keep multiple experts resident. The serving cost implications deserve more emphasis.

3. Comparison fairness around embeddings. For NLP, “the quantization experts’ embedding layer is left in full precision” and “MoQE uses the pre-existing embedding layer of the original full-precision model”. It’s unclear whether baselines also keep embeddings FP; if not, MoQE might enjoy an unfair FP advantage in the input stack.

**Questions:**

1. Generalization beyond PPL. Most NLP results are perplexity-only on C4/Wiki/WebText . Could you add zero-shot/GLUE/MMLU-style tasks (or CV transfer beyond ImageNet) to validate that MoQE’s gains transfer to tasks rather than only PPL?

2. Router label generation & data leakage. Router labels are assigned by “the expert that yields the lowest loss” per sample (training uses the training set). Is labeling performed on the same data used to train the router? If so, can you provide an evaluation where labels are computed on a held-out set to avoid teacher-student leakage?

---

> ### Author Response · Authors · 2025-11-16
> **Thank you for your valuable review**
>
> Below is our point-by-point response to the Weaknesses (W) and Questions (Q) you raised:
> # Regarding W1&Q1 : Evaluation scope (datasets & metrics) is narrow for NLP
> We conducted a series of new downstream task evaluations, using the Qwen-0.6B, Qwen-1.7B,Llama-3B, Qwen-4B,Qwen-14B model, comparing the performance of MoQE  with the best single quantization expert in downstream tasks (higher score is better).
> The new experimental results are as follows:
> | Quantization method (4bit) | MMLU (0.6B) | MMLU (1.7B) | MMLU (4B) | MMLU (14B) |
> |----------------------------|-------------|-------------|-----------|------------|
> | FP16                       | 47.1        | 60.0        | 69.7      | 78.5       |
> | AWQ                        | 43.1        | 53.9        | 66.0      | 76.3       |
> | GPTQ                       | 40.0        | 52.8        | 65.8      | 75.9       |
> | K-Quants                   | 32.2        | 41.8        | 62.6      | 68.4       |
> | SmoothQuant                | 30.8        | 44.1        | 59.2      | 71.3       |
> | imatrix                    | 38.4        | 47.5        | 65.1      | 70.6       |
> | MOQE                       | 45.5        | 55.9        | 68.3      | 77.9       |
>
> | Quantization method (4bit) | GLUE (0.6B) | GLUE (1.7B) | GLUE (4B) | GLUE (14B) |
> |----------------------------|-------------|-------------|-----------|------------|
> | FP16                       | 77.0        | 87.56       | 90.5      | 92.5       |
> | AWQ                        | 70.4        | 78.6        | 85.7      | 89.8       |
> | GPTQ                       | 65.4        | 77.0        | 85.5      | 89.3       |
> | K-Quants                   | 52.7        | 60.9        | 81.2      | 80.4       |
> | SmoothQuant                | 50.4        | 64.2        | 76.8      | 83.9       |
> | imatrix                    | 62.8        | 69.1        | 84.6      | 83.0       |
> | MOQE                       | 72.5        | 80.5        | 88.1      | 91.8       |
>
> | Quantization method(8bit) | GLUE (1.7B) | GLUE (0.6B) |MMLU (4B) | MMLU (Llama-3B) |
> |----------------------------|-------------|-----------|---|---|
> | FP16                       | 87.56       | 77.0      |69.7 | 63.4 |
> | AWQ                        | 87.4        | 76.8      |69.5 | 63.1 |
> | GPTQ                       | 87.4        | 75.5      |59.9 | 61.7 |
> | K-Quants                   | 83.5        |  75.1    |64.2 | 60.4 |
> | SmoothQuant                | 86.99       | 75.9      |69.3 | 62.3 |
> | imatrix                    | 85.8        |  75.4     |69.4 | 61.4 |
> | MOQE                       | 87.5        | 76.8      |69.7 | 63.3 |
>
> As the new MMLU/GLUE results show , MoQE  consistently outperforms the best single expert across all models. This confirms our gains transfer to downstream tasks and narrow the gap to FP16.
> # Regarding W2:Cost/footprint trade-offs underplayed.
> The memory footprint data for the 4-expert configuration in our experiments (see Table 8 and Table 9) best illustrates this point.
> | Model | Resource type | MoQE | Single quantization model | Total resources | Resource increment |
> | :--- | :--- | :--- | :--- | :--- | :--- |
> | Qwen-1.7B | GPU VRAM | ~7.86 GB | ~7.64 GB | 80G | +0.22GB (+0.2%） |
> | | CPU RAM | ~5.17 GB | ~0 | 128G~2T | +5.17 GB (+0.2%-4%） |
> | LLAMA-3B | GPU VRAM | ~10.43 GB | ~10.17 GB | 80G | +0.26 GB (+0.3%) |
> | | CPU RAM | ~9.43 GB | ~0 | 128G~2T | +9.43 GB(+0.4%-7%） |
> | Qwen-4B | GPU VRAM | ~14.76 GB | ~14.30 GB | 80G | +0.46 GB (+0.5%） |
> | | CPU RAM | ~12.23 GB | ~0 | 128G~2T | +12.23 GB(+0.6%-9%） |
>
> As shown in Table, MoQE compared to the single quantization model baseline, the VRAM increase is very minimal. Current mainstream server memory sizes are usually greater than 128GB, so we believe that exchanging acceptable CPU RAM for zero VRAM growth and significant model precision improvement is a very pragmatic and efficient deployment strategy.
> # Regarding Q2:Router label generation&data leakage.
> Our experiments strictly distinguished between data used for training and unseen data used for evaluation. The dataset we used when training the routing model is a separate dataset; regarding key data evaluation, including the content in the paper and all experimental content, it was conducted on a completely independent validation dataset.
> # Regarding W3:Comparison fairness around embeddings.
> All baselines compared in the paper also kept the embedding layer in full precision during our evaluation. Therefore, our comparison is completely fair. At the same time, existing LLM PTQ algorithms usually do not quantize the embedding layer. This is a common practice because the embedding layer itself is sparse , and quantizing storage does not offer significant benefits for it. Secondly, because it does not involve computation , there is no need to quantize it. More importantly, the information it stores has high precision requirements, and quantization would bring enormous errors. Therefore, most quantization methods do not quantize the embedding layer.

---

> ### Author Response · Authors · 2025-11-22
> **Follow up on rebuttal**
>
> Dear Reviewer sasr,
>
> We sincerely appreciate your valuable comments! We have provided detailed responses to each of your concerns and made the necessary revisions. We kindly ask if you are satisfied with our revisions and if you have any further concerns or feedback. We would be more than happy to continue the discussion and address any additional points you may have.
>
> Thank you for your time and consideration.
>
> Best regards,
>
> The Authors

---

> ### Author Response · Authors · 2025-11-25
> **Follow up on rebuttal**
>
> Dear Reviewer sasr,
>
> We hope our previous responses have adequately addressed your concerns and clarified the points you raised. As the discussion phase is coming to a close, we would greatly appreciate it if you could take a moment to review our rebuttal.
>
> We welcome any further discussions if you have additional questions. Thank you for your time and effort in reviewing our paper.
>
> Best regards,The Authors

---

> ### Author Response · Authors · 2025-11-27
> **Follow up on rebuttal**
>
> Dear Reviewer sasr,
>
> As the discussion period draws to a close, we wanted to express our sincere gratitude for your exceptionally thorough and constructive feedback.
>
> We truly value the time and effort you dedicated to helping us improve this work.We believe this has effectively addressed the main issues that affected your initial rating. As this potential defect has been resolved, we sincerely hope that you will consider updating your score.
>
> Sincerely, Authors

---

> ### Author Response · Authors · 2025-11-28
> **Follow up on rebuttal**
>
> Dear Reviewer sasr,
>
> We sincerely wish you continued success in your recent work and hope you had a pleasant Thanksgiving holiday.
>
> We apologize for disturbing you. We have prepared a comprehensive response that thoroughly addresses all the concerns you raised. As this discussion phase concludes, we would appreciate it if you could confirm whether our explanations have sufficiently clarified your remaining questions. We sincerely hope that you will view our revisions favorably and, of course, remain available to address any additional inquiries you may have.
>
> we respectfully wonder if you might see fit to upgrade your score.
>
> Sincerely, Authors

---

### Official Review · Reviewer_QQqJ · 2025-11-04

**Soundness:** 3
**Presentation:** 3
**Contribution:** 2
**Rating:** 4
**Confidence:** 4

**Summary:**

This paper proposes Mixture of Quantization Experts (MoQE), a novel post-training quantization framework that enhances model accuracy using a Mixture-of-Experts (MoE) architecture. Instead of relying on a single quantized model, MoQE integrates multiple quantization variants of the same full-precision model as “quantization experts” and uses a lightweight router model to dynamically assign each input to the most suitable expert. Extensive experiments on ResNet, MobileNet, LLaMA, and Qwen across datasets such as ImageNet, WikiText, C4, and OpenWebText show that MoQE consistently outperforms individual quantization methods (e.g., GPTQ, SmoothQuant, AWQ) at both Int8 and Int4 bitwidths.

**Strengths:**

1) Experiments are comprehensive, covering ResNet, MobileNet, LLaMA, and Qwen models across diverse datasets (ImageNet, WikiText, C4, OpenWebText).
2) The paper is generally well written and clearly structured, with extensive figures, tables, and methodological details.
3) The proposed MoQE achieves consistent performance gains over state-of-the-art quantization baseline.

**Weaknesses:**

1) The paper provides limited theoretical justification for why dynamic routing among quantized experts improves accuracy. A formal analysis of router convergence, expert diversity, or error bounds would strengthen the contribution.
2) While the idea is interesting, it may be perceived as a straightforward combination of two existing ideas, quantization and MoE routing, rather than a fundamentally new theoretical contribution.
3) The paper could be improved by including real-world deployment results, and additional baselines such as adaptive or ensemble quantization methods.
4) The work would benefit from a deeper discussion on when MoQE might fail, e.g., highly homogeneous data distributions or limited quantization diversity, and how the approach could adapt in such scenarios.
5) The performance improvements reported in Tables 1–3 are relatively small compared to the baselines.

**Questions:**

1) The framework relies heavily on the assumption that different quantization methods introduce complementary biases. Could the authors provide quantitative evidence (e.g., disagreement metrics or diversity indices) showing that these experts are sufficiently distinct to justify a MoE-style combination?
2) The experiments mainly cover models up to Qwen-4B and LLaMA-3B. How would the MoQE system scale for >10B-parameter models, where router overhead and expert management may become non-trivial?
3) The work provides intuitive explanations but lacks a formal analysis of routing optimality or error bounds. Could the authors elaborate on theoretical insights explaining why routing among quantized experts yields lower loss than single-expert quantization?

---

> ### Author Response · Authors · 2025-11-21
> **Thank you for your valuable review.Below is our point-by-point response to the Weaknesses (W) and Questions (Q) you raised.**
>
> # W1 & Q3
> We have updated the paper to present the theoretical upper bound of the MoQE system's loss (highlighted in red in the revised version, with the proof provided in the AppendixA.8). This theorem demonstrates that the loss upper bound of the MoQE system is no bigger than that of a single quantization expert.This provides a detailed explanation of why routing between quantization experts yields lower loss compared to single quantization expert.
> # W2 & Q1
> As demonstrated in our response to Reviewer mBvS, the loss degradation for the same sample varies significantly across different quantization models. Consequently, different quantization methods exhibit distinct biases toward the same sample. Liu et al. (2025) [1] established that this bias is inherent to quantization algorithms and is unavoidable.
>
> Our theoretical analysis confirms that the biases of quantization models with different parameters are distinct. Our approach is an algorithmic innovation specifically designed to exploit these biases to optimize system performance. Instead of the traditional approach of merely combining different precisions [2][3][4], we combine distinct quantization algorithms and route data to the most suitable node.
>
> Therefore, MoQE performs routing at the whole-model level rather than constructing a separate MoE layer. This design choice is a necessary response to the inherent bias problem, rather than a trivial combination of two existing ideas. Whether based on observation, theoretical analysis, or methodological design, our work presents a novel contribution that is absent in current literature.
> # W3&Q2
> We conducted a detailed latency experiment on the Qwen-14B model. The results demonstrate that MoQE maintains high efficiency on large-scale models:
>
> Table 1: Inference Overhead Analysis
> | Metric | Time (ms) | Percentage |
> | :--- | ---: | ---: |
> | Expert Inference Time | 37,236 | - |
> | EIT | 545 | 1.46% |
> | Switching Overhead | 1,110 | 2.9% |
> |Total Overhead | 1,655 | 4.36% |
>
> The total additional overhead (4.36%) remains below 5%. This compellingly demonstrates that as the model scales from 0.6B to 14B parameters, the aggregate overhead becomes negligible relative to the substantial computational cost of inference.
> Table 2: MMLU Performance (Qwen 0.6B–14B)
> | method (4bit) | MMLU (0.6B) | MMLU (1.7B) | MMLU (4B) | MMLU (14B) |
> |:--------------|:------------|:------------|:----------|:-----------|
> | FP16          | 47.1        | 60          | 69.7      | 78.5       |
> | AWQ           | 43.1        | 53.9        | 66        | 76.3       |
> | GPTQ          | 40          | 52.8        | 65.8      | 75.9       |
> | K-Quants      | 32.2        | 41.8        | 62.6      | 68.4       |
> | SmoothQuant   | 30.8        | 44.1        | 59.2      | 71.3       |
> | imatrix       | 38.4        | 47.5        | 65.1      | 70.6       |
> | MOQE          | 45.5        | 55.9        | 68.3      | 77.9       |
> # W4
> In the revised version of paper, we provide a complete proof of Theorem 1. This theorem states that when a single quantization expert outperforms other models in the all dataset, the performance of MoQE is equivalent to that of that expert.
> # W5
> Table 2 presents the MMLU scores of MoQE, where the MoQE (Int4) score improves by an average of 2 points and a maximum of 2.4 points. In published papers [5][6][7], the PPL decreased by 0.7, 0.6, and 0.5, respectively; these papers have already been accepted by ICLR. In contrast, MoQE achieves a reduction of approximately 1.0. The performance improvement of our method exceeds the performance improvements reported in these articles; therefore, our performance improvement is not small.
>
> # Reference
> [1] Liu, B., et al. (2025). Understanding the Unfairness in Network Quantization. Proceedings of the 42nd International Conference on Machine Learning (ICML), PMLR 267:39106-39125.
>
> [2] Kim, Y. J., et al. (2023). Mixture of Quantized Experts (MoQE): Complementary Effect of Low-bit Quantization and Robustness. arXiv preprint arXiv:2310.02410.
>
> [3] Chitty-Venkata, K. T., et al. (2025). MoPEQ: Mixture of Mixed Precision Quantized Experts. Proceedings of the IEEE/CVF International Conference on Computer Vision (ICCV) Workshops, pp. 4023–4032.
>
> [4] Fu, Z., et al. (2025). EAQuant: Enhancing Post-Training Quantization for MoE Models via Expert-Aware Optimization. arXiv preprint arXiv:2506.13329.
>
> [5] Liu, Z., et al. (2025). SpinQuant: LLM Quantization with Learned Rotations. Proceedings of the International Conference on Learning Representations (ICLR), 2025.
>
> [6] Saxena, U., et al. (2024). ResQ: Mixed-Precision Quantization of Large Language Models with Low-Rank Residuals. Proceedings of the International Conference on Learning Representations (ICLR), 2025.
>
> [7] Shao, W., et al. (2024). OmniQuant: Omnidirectionally Calibrated Quantization for Large Language Models. Proceedings of the International Conference on Learning Representations (ICLR), 2024.

---

> > ### Author Response · Authors · 2025-11-22
> > **Follow up on rebuttal**
> >
> > Dear Reviewer QQqJ,
> >
> > We sincerely appreciate your insightful comments and the time you have taken to review our work. In response, we have carefully addressed each of your concerns and made the necessary revisions. We kindly ask if these revisions meet your expectations and if there are any further concerns or feedback you would like to discuss. We are more than happy to continue the conversation and address any additional points you may have.
> >
> > Thank you once again for your time and thoughtful consideration.
> >
> > Best regards, The Authors

---

> ### Author Response · Authors · 2025-11-25
> **Follow up on rebuttal**
>
> Dear Reviewer QQqJ,
>
> We apologize for disturbing you. As the discussion deadline approaches, we are writing to follow up to ensure that our previous response and revisions have fully addressed your concerns.
>
> We believe that these improvements have significantly strengthened the quality of the paper. We would be extremely grateful if you could take a moment to review our reply and re-evaluate our work. We are joyous to engage in further discussion if needed.
>
> Best regards, The Authors

---

> ### Author Response · Authors · 2025-11-27
> **Follow up on rebuttal**
>
> Dear Reviewer QQqJ,
>
> Thank you again for your constructive suggestions. Your feedback has significantly improved the quality of our article.
>
> We have provided a detailed response addressing your concerns.As the discussion is coming to an end, we would like to kindly confirm whether our response has clarified your doubts. We hope you will reconsider our improvements and are ready to answer any other questions you may have. We respectfully wonder if you might consider reconsidering your score to reflect the enhanced quality of the paper.
>
> Sincerely, Authors

---

> ### Author Response · Authors · 2025-11-28
> **Follow up on rebuttal**
>
> Dear Reviewer QQqJ,
>
> We sincerely wish you continued success in your recent work and hope you had a pleasant Thanksgiving holiday.
>
> We have prepared a comprehensive response that thoroughly addresses all the concerns you raised. As this discussion phase concludes, we would appreciate it if you could confirm whether our explanations have sufficiently clarified your remaining questions. We sincerely hope that you will view our revisions favorably and, of course, remain available to address any additional inquiries you may have.
>
> we respectfully wonder if you might see fit to upgrade your score.
>
> Sincerely, Authors

---

### Author Response · Authors · 2025-12-03
**Summary**

Dear  Area Chair

We are grateful for the constructive comments from the three reviewers and also thank you for your contribution to the community. We understand the challenges brought by the recent review reset. To assist in your evaluation, we have summarized the significant improvements made during the discussion period to address potential concerns:

## Innovation
This paper presents an algorithmic system innovation based on the inherent property of quantization models that "the loss of the same sample varies significantly under different quantization models" [1]. This inherent phenomenon has been fully proven by experiments and theory (Reviewer mBvS Q1, Reviewer QQqJ Q1). We further analyzed the relationship between unfairness and quantization methods as well as quantization precision in the paper (Reviewer mBvS Q2, Q3).

Our method is an algorithmic innovation system aimed at leveraging these deviations to optimize system performance. Unlike traditional methods that simply combine different precisions [2][3][4], we combine different quantization algorithms and route data to the most suitable nodes. This design choice is a necessary response to the inherent deviation problem, not a simple combination of two existing ideas. Whether based on observation, theoretical analysis, or method design, our work presents novel contributions not present in current literature. (Reviewer QQqJ W2)

## Theoretical Analysis:
We have updated the paper to provide the theoretical upper bound of the loss for the MoQE system (highlighted in red in the paper, proof in Appendix A.3). This theorem shows that the loss upper bound of the MoQE system is not greater than the loss upper bound of a single quantization expert. This explains in detail why routing among multiple quantization experts results in lower loss than using a single quantization model. (Reviewer QQqJ W1 & Q3)

Theorem 1 states that when a single quantization expert outperforms other models on all datasets, the performance of MoQE is the same as that expert's performance (Reviewer QQqJ W4). However, in real-world scenarios, we find that this situation rarely occurs.

## Algorithm Details
We clarified that the embedding layer is not quantized because all baseline methods compared in this paper maintain full precision for the embedding layer during evaluation. Therefore, this algorithmic detail is completely fair. We  also clarified in the paper that in the algorithm design, the experiments strictly distinguish between data used for training and unseen data used for evaluation. The dataset used to train the routing model is independent; whereas key data evaluations, including paper content and all experimental content, are conducted on a completely independent validation dataset. (Reviewer sasr Q2)

We analyzed the idea of selecting different quantization experts for the prefill and decode stages in LLMs. We call this method "two-stage routing," and demonstrated through experiments that two-stage routing indeed yields lower loss. However, the "two-stage routing" strategy faces the problem of KV cache incompatibility. This would introduce huge computational overhead and latency; such massive latency is unacceptable to us, but we believe this is indeed a good idea worth serious research.

## Experimental Results
We conducted more experiments specifically on the effectiveness of the MoQE system. We provided downstream experiments for MoQE, measuring metrics such as MMLU and GLUE, all of which proved MoQE's superiority under different metrics, especially demonstrating greater advantages in real-world metrics (Reviewer QQqJ W5, Reviewer sasr W1 & Q1). At the same time, compared to previous works [5][6][7], our work has advantages in the level of performance improvement. (Reviewer QQqJ W5)

Regarding LLM performance at a larger scale, we added the model performance of Qwen-14B. This experiment indicates that on large-scale models, MoQE can maintain high efficiency in terms of both model effectiveness and running speed. (Reviewer QQqJ W3 & Q2)

we supplemented the memory resource usage of MoQE. Experiments show that compared to the single quantization model baseline, the increase in GPU memory (VRAM) for MoQE is very small. Currently, the memory capacity of mainstream servers is usually greater than 128GB, so we believe that trading acceptable CPU memory usage for a slight increase in GPU memory and a significant improvement in model accuracy is a very pragmatic and efficient method. (Reviewer sasr W2)

## Conclusion
The reviewers provided valuable comments on our innovation, algorithm design, theoretical analysis, and experiments. We have replied point-by-point and improved the paper. Reviewer mBvS  expressed that our response completely resolved all his confusions, and other reviewers also expressed agreement and raised no additional objections.

Best regards, The Authors

---

> ### Author Response · Authors · 2025-12-03
> **References**
>
> # Reference
>
> [1] Liu, B., et al. (2025). Understanding the Unfairness in Network Quantization. Proceedings of the 42nd International Conference on Machine Learning (ICML), PMLR 267:39106-39125.
>
> [2] Kim, Y. J., et al. (2023). Mixture of Quantized Experts (MoQE): Complementary Effect of Low-bit Quantization and Robustness. arXiv preprint arXiv:2310.02410.
>
> [3] Chitty-Venkata, K. T., et al. (2025). MoPEQ: Mixture of Mixed Precision Quantized Experts. Proceedings of the IEEE/CVF International Conference on Computer Vision (ICCV) Workshops, pp. 4023–4032.
>
> [4] Fu, Z., et al. (2025). EAQuant: Enhancing Post-Training Quantization for MoE Models via Expert-Aware Optimization. arXiv preprint arXiv:2506.13329.
>
> [5] Liu, Z., et al. (2025). SpinQuant: LLM Quantization with Learned Rotations. Proceedings of the International Conference on Learning Representations (ICLR), 2025.
>
> [6] Saxena, U., et al. (2024). ResQ: Mixed-Precision Quantization of Large Language Models with Low-Rank Residuals. Proceedings of the International Conference on Learning Representations (ICLR), 2025.
>
> [7] Shao, W., et al. (2024). OmniQuant: Omnidirectionally Calibrated Quantization for Large Language Models. Proceedings of the International Conference on Learning Representations (ICLR), 2024.

---

### Note · Authors · 2026-01-26

I have read and agree with the venue's withdrawal policy on behalf of myself and my co-authors.

---

### Meta-Review · Area_Chair_xTAP · 2026-01-04

**Summary:**

This paper proposes a Mixture-of-Quantization-Experts (MoQE) approach that combines PTQ with MoE-style routing to improve model accuracy without increasing GPU memory footprint. By maintaining multiple quantized variants of a model and routing inputs among them, the method reports consistent accuracy gains across several vision and language benchmarks. Reviewers found the idea intuitive and empirically promising, but raised concerns regarding novelty, deployment overhead, and the claimed efficiency tradeoffs.

**Reviewer Concerns:**

The rebuttal addressed several presentation and analysis issues, including adding theoretical justification, expanding downstream task evaluation, and clarifying latency and memory measurements under the authors' experiment setup. However, two major concerns remain unresolved.

First, the contribution is largely a straightforward composition of existing ideas, e.g., post-training quantization and MoE-style routing, which limits conceptual novelty. As a result, the acceptance of the paper lies on strong empirical validation of the claimed deployment tradeoffs.

Second, and more critically, the paper does not convincingly validate its practicality for latency-sensitive autoregressive inference. Maintaining multiple quantized models force some models to be placed on CPU memory and introduces additional complexity not present in standard quantized models. In realistic autoregressive inference settings, routing may reduce batching efficiency and trigger frequent cache misses or model swap-in/out over PCIe, which can significantly impact TTFT and TBT latency. While the paper frames the method as exchanging CPU RAM for zero VRAM growth and improved accuracy, the evaluation does not demonstrate that this tradeoff preserves latency or SLOs under autoregressive decoding. In addition, the learned router relies on calibration data, raising concerns about potential bias or overfitting under distribution shift.

Taken together, these issues make the practical value of the approach not convincingly established.

**Reviewer Scores:**

Reviewer mBvS: Likely unchanged (6); concerns about novelty and deployment overhead remain despite added analysis.

Reviewer sasr: Likely unchanged (6); evaluation improved, but practicality under inference workloads remains unclear.

Reviewer QQqJ: Likely unchanged (4); doubt about real-world deployment results and tradeoffs is not fully addressed.

---

### Decision · Program_Chairs · 2026-01-26

Reject